# Bridging Functional and Representational Similarity via *Usable* Information

**Antonio Almudévar** [1]  **Alfonso Ortega** [1]

## Abstract

We present a unified framework for quantifying the similarity between representations through the lens of *usable* information, offering a rigorous theoretical and empirical synthesis across three key dimensions. First, addressing functional similarity, we establish a formal link between stitching performance and conditional mutual information. We further reveal that stitching is inherently asymmetric, demonstrating that robust functional comparison necessitates a bidirectional analysis rather than a unidirectional mapping. Second, concerning representational similarity, we find that reconstruction-based metrics and standard tools (e.g., CKA, RSA) act as estimators of usable information under specific constraints. Crucially, we show that similarity is relative to the capacity of the predictive family: representations that appear distinct to a rigid observer may be identical to a more expressive one. Third, we demonstrate that representational similarity is sufficient but not necessary for functional similarity. We unify these concepts through a task-granularity hierarchy: similarity on a complex task guarantees similarity on any coarser derivative, establishing representational similarity as the limit of maximum granularity: input reconstruction.

## 1. Introduction

The success of deep learning stems largely from the internal representations learned by neural networks. These representations encode structured and often compressed information about the input, driving strong performance across diverse tasks (Bengio et al., 2013; Tishby & Zaslavsky, 2015). Comparing representations has emerged as a central question—not only in deep learning, but also in neuroscience and cognitive science (Kriegeskorte et al., 2008b; Jacoby & McDermott, 2017; Sucholutsky et al., 2023)—as it reveals how different systems process and organize information. These comparisons address fundamental questions such as: (i) whether different models capture the same structures (Li et al., 2015), (ii) whether knowledge can be transferred across models (Yosinski et al., 2014), and (iii) whether performance differences stem from the representations or from the task heads (Alain & Bengio, 2016).

However, before developing or applying metrics to compare representations, it is essential to clarify what exactly should be compared—in other words, what it means for two representations to be similar or dissimilar. The literature typically distinguishes between two complementary notions of similarity: (i) *functional similarity*, which assesses whether they support the same downstream behavior—i.e., whether their encoded information is equivalently usable for predicting a given output (Lenc & Vedaldi, 2015; Ding et al., 2021); and (ii) *representational similarity*, which assesses whether two representations encode input information in the same way—i.e., whether they preserve equivalent relational or informational structure (Kriegeskorte et al., 2008a). Despite their intuitive connection, these notions remain loosely defined and have been studied largely in isolation.

The lack of a unified framework has disconnected functional methods (e.g., stitching) from geometric metrics (e.g., CKA, RSA). We resolve this by grounding both in the theory of *usable information* (Xu et al., 2020), defining similarity as relative to the computational constraints of a *predictive family*. Our main contributions are:

1. **A Unified Framework:** We theoretically link model stitching to usable conditional mutual information, showing that functional similarity is defined relative to the capacity of the predictive family.

2. **Reinterpreting Standard Metrics:** We empirically find that standard metrics like CKA, RSA, and Procrustes are not arbitrary geometric measures, but specific estimators of usable information.

3. **The Hierarchy of Similarity:** We show representational similarity is sufficient, but not necessary, for functional similarity. We bridge this by establishing similarity is monotonic with respect to task granularity, identifying representational similarity as the limit of maximum granularity: input reconstruction.

[1]ViVoLab, Aragón Institute for Engineering Research (I3A), University of Zaragoza, Zaragoza, Spain. Correspondence to: Antonio Almudévar <almudevar@unizar.es>.

*Proceedings of the 43rd International Conference on Machine Learning*, Seoul, South Korea. PMLR 306, 2026. Copyright 2026 by the author(s).

## 2. Preliminaries

### 2.1. Usable Information Theory

Classical information theory quantifies the total statistical dependence between random variables through the mutual information $I(Y; Z)$, which measures how much knowledge of $Z \in \mathcal{Z}$ reduces the uncertainty about $Y \in \mathcal{Y}$ (Reza, 1994; Cover, 1999). However, in learning systems, not all of this information is necessarily usable by a downstream model. A representation may encode fine-grained details of $Y$ that, while statistically present, cannot be effectively exploited by a predictor of limited capacity (Achille & Soatto, 2018). In other words, the amount of information that is usable by a learner depends not only on the representation itself but also on the inductive biases and expressiveness of the predictive model (Saxe et al., 2019).

*Usable information theory* (Xu et al., 2020; Ethayarajh et al., 2022) formalizes this by restricting information flow to a *predictive family* $\mathcal{V}$. We define the $\mathcal{V}$-*conditional entropy* $H_\mathcal{V}(Y \mid Z)$ and the marginal $\mathcal{V}$-*entropy* $H_\mathcal{V}(Y \mid \varnothing)$ as the minimum cross-entropy loss achievable by a predictor in $\mathcal{V}$:

$$H_\mathcal{V}(Y \mid Z) = \inf_{f \in \mathcal{V}} \mathbb{E}_{p(y,z)}\big[ - \log f[z](y) \big] \qquad (1)$$

$$H_\mathcal{V}(Y \mid \varnothing) = \inf_{f \in \mathcal{V}} \mathbb{E}_{p(y)}\big[ - \log f[\varnothing](y) \big] \qquad (2)$$

where $f$ is a function $\mathcal{Z} \cup \{\varnothing\} \to \mathcal{P}(\mathcal{Y})$, so $f[z] \in \mathcal{P}(\mathcal{Y})$ is a probability measure on $\mathcal{Y}$ chosen based on $z$; and $f[z](y) \in \mathbb{R}$ is the value of the density evaluated at $y \in \mathcal{Y}$.

We define the *usable information* from $Z$ to $Y$ as the reduction in uncertainty provided by $Z$ relative to $\mathcal{V}$:

$$I_\mathcal{V}(Z \to Y) = H_\mathcal{V}(Y \mid \varnothing) - H_\mathcal{V}(Y \mid Z). \qquad (3)$$

Unlike Shannon mutual information, usable information is directional: $I_\mathcal{V}(Z \to Y) \neq I_\mathcal{V}(Y \to Z)$. For instance, if $\mathcal{V}$ contains only linear predictors, $I_\mathcal{V}(Z \to Y)$ quantifies how much of the information in $Z$ is linearly usable for predicting $Y$. A representation that encodes $Y$ in a highly nonlinear form may exhibit high total information $I(Y; Z)$ but low usable information $I_\mathcal{V}(Z \to Y)$.

Finally, the *usable conditional information* quantifies the information in $Z$ about $Y$ usable beyond a variable $W$. It is defined as the reduction in $\mathcal{V}$-entropy achieved by adding $Z$ to the context $W$:

$$I_\mathcal{V}(Z \to Y \mid W) = H_\mathcal{V}(Y \mid W) - H_\mathcal{V}(Y \mid Z, W). \qquad (4)$$

This quantity measures the additional functional value of $Z$ given $W$. When $\mathcal{V}$ is unrestricted, these metrics recover standard Shannon definitions. Thus, usable information bridges statistical and functional perspectives, precisely quantifying which aspects of a representation can be exploited by downstream models.

### 2.2. Functional Similarity

Functional similarity measures whether two representations support the same downstream behaviors, regardless of how information is encoded internally. Representations are functionally similar if predictors trained on one operate effectively on the other, potentially via a transformation layer aligning feature spaces.

A common framework is *model stitching* (Lenc & Vedaldi, 2015; Bansal et al., 2021; Hernandez et al., 2023; Pan et al., 2023), combining one encoder with another's decoder to evaluate how well their intermediate representations interface. High stitching performance implies comparable task-relevant information. Recent work on *relative representations* (Moschella et al., 2022) posits that functional similarity stems from preserving topological relations (distances and angles) rather than absolute coordinates.

Related approaches include *linear probes* (Alain & Bengio, 2016; Hewitt & Liang, 2019), *transfer evaluations* (Zamir et al., 2018; Kornblith et al., 2019), and *functional alignment* methods that measure performance consistency across models (Geirhos et al., 2020).

### 2.3. Representational Similarity

Representational similarity metrics quantify the alignment between two sets of representations, $Z_A \in \mathbb{R}^{n \times d_A}$ and $Z_B \in \mathbb{R}^{n \times d_B}$. These methods generally fall into two categories: those that explicitly align the feature spaces and those that compare intrinsic structures (Ding et al., 2021).

Alignment-based methods minimize element-wise distance. *Linear Regression* (Li et al., 2015) finds the optimal linear mapping, providing invariance to invertible linear transformations. *Procrustes Analysis* (Schönemann, 1966), restricts the mapping to an orthogonal rotation. Thus, Procrustes is invariant strictly to *orthogonal transformations*.

In contrast, structural methods compare geometric properties without direct feature alignment, typically sharing invariance to *orthogonal transformations* and *isotropic scaling*. Key approaches include *Canonical Correlation Analysis* (CCA) and its extensions (Raghu et al., 2017; Morcos et al., 2018), which align subspaces via maximizing linear correlations; *Centered Kernel Alignment* (CKA; Kornblith et al., 2019), which compares similarity kernels via the Hilbert-Schmidt Independence Criterion (Gretton et al., 2005); and *Representational Similarity Analysis* (RSA; Kriegeskorte et al., 2008a), which assesses the rank-ordering of pairwise distance matrices.

As elaborated in Section 4, these invariances are not merely geometric details; in the context of usable information, they implicitly define the specific predictive family $\mathcal{V}$ used to compare the representations.

# 3. *What* information is encoded

In this section, we focus on defining similarity[1] solely in terms of *what* information the representations encode. Crucially, since all our definitions are grounded in information theory, we abstract away from *how* this information is represented—a question addressed later in Section 4. The main results of this section are derived in Appendix A

## 3.1. Representations and Markov Blankets

In machine learning, we assume a dataset of *inputs* $x \in \mathcal{X}$ and corresponding *tasks* $y \in \mathcal{Y}$. The goal is to predict $\hat{y}$ such that it closely matches $y$. To achieve this, systems rely on an intermediate *representation* $z \in \mathcal{Z}$, a compressed version of $x$. We formalize this below:

**Definition 3.1** (Representation). A variable $Z$ is a *representation* of input $X$ if $Z$ is a stochastic function of $X$, fully characterized by $p_\theta(z \mid x)$.

The distribution $p_\theta(z \mid x)$ is the *encoder* (parameterized by $\theta$). Subsequently, the prediction $\hat{y}$ is generated by a *task head* $q_\phi(\hat{y} \mid z)$ (parameterized by $\phi$).

We introduce the *Markov blanket* to formalize when one variable screens off another from a third (Pearl, 2014).

**Definition 3.2** (Markov blanket). Let $A$, $B$, and $C$ be random variables. $B$ is a *Markov blanket* between $A$ and $C$ if the Markov chain $A \leftrightarrow B \leftrightarrow C$ holds. Equivalently, $B$ contains all the information about $C$ present in $A$; that is, $I(C; A \mid B) = 0$.

## 3.2. Functional Similarity

Informally, representations $Z_1$ and $Z_2$ are *functionally similar* if they yield similar predictions for a task $y$. Since this notion is inherently ambiguous, we formalize it using the Markov blanket concept.

**Definition 3.3** (Functional Similarity). Let $Z_1$ and $Z_2$ be representations and $Y$ a task. $Z_1$ and $Z_2$ are *functionally similar* w.r.t. $Y$ if $Z_1$ forms a Markov blanket between $Z_2$ and $Y$, and vice versa. Equivalently:

$$I(Z_2; Y \mid Z_1) = I(Z_1; Y \mid Z_2) = 0. \tag{5}$$

This implies $Z_1$ and $Z_2$ contain identical information about $Y$. Note that this does not require predictions from arbitrary task heads $q_{\phi_1}$ and $q_{\phi_2}$ to be identical, as outputs depend on the head's capacity. Rather, it implies that given *perfect* heads—capable of extracting all available information—the predictions would be equivalent.

---

[1]With a slight abuse of notation to fit established literature nomenclature, we use the term *similarity* throughout this section to denote a state of *perfect equivalence* (i.e., zero conditional mutual information). We bridge this strict theoretical condition to practical, continuous metrics in Section 4.

**Model Stitching**   A common method for assessing similarity is *model stitching*, which maps $z_1$ to $z_2$ via a *stitcher* $s \in \mathcal{S}$ to minimize $\text{CE}\big(p(y, x), q_{\phi_2}(\hat{y} \mid s(z_1))\big)$, where $q_{\phi_2}$ denotes the task head that predicts $y$ from $z_2$.

We argue that stitching is not merely heuristic but is fundamentally defined by the Markov blanket relationship.

**Definition 3.4** (Perfect Stitchability). $Z_1$ is *perfectly stitchable* into $Z_2$ given a predictor $q_{\phi_2}$ if there exists a stitcher $s$ such that the cross-entropy (CE) losses are identical:

$$\text{CE}\big(p(y, x), q_{\phi_2}(\hat{y} \mid z_2)\big) = \text{CE}\big(p(y, x), q_{\phi_2}(\hat{y} \mid s(z_1))\big). \tag{6}$$

**Proposition 3.5** (Markov Blanket–Stitchability Equivalence under Optimality). *Let $q_{\phi_2}$ be a Bayes-optimal predictor for $Z_2$. Then, $Z_1$ is a Markov blanket for $Y$ relative to $Z_2$ if and only if $Z_1$ is perfectly stitchable into $Z_2$.*

Intuitively, if $Z_1$ holds all of $Z_2$'s information about $Y$, a model can extract it. Conversely, by the *Data Processing Inequality* (Cover, 1999), a deterministic stitcher cannot generate new information not already in $Z_1$. This leads to the direct link between stitching and functional similarity:

**Corollary 3.6** (Functional Similarity–Stitchability Equivalence). *$Z_1$ and $Z_2$ are functionally similar w.r.t. $Y$ if and only if $Z_1$ is perfectly stitchable into $Z_2$ and vice versa.*

Thus, establishing functional similarity requires identifying two stitchers: one mapping $Z_1 \to Z_2$ and another $Z_2 \to Z_1$. Consider, for instance, a neural network layer transforming input $Z_{in}$ to output $Z_{out}$. A forward stitcher exists trivially (the layer itself), meaning $Z_{in}$ is a Markov blanket between $Z_{out}$ and $Y$. However, a reverse stitcher likely fails due to information loss in non-invertible layers (Jacobsen et al., 2018). Consequently, they are not functionally similar, demonstrating that stitchability is inherently directional. While this asymmetry has been previously observed empirically (Bansal et al., 2021), our framework formally grounds it as a mathematical inevitability of the Markov blanket condition.

> **Takeaway.** Assessing functional similarity requires two well-performing stitchers: one from $Z_1$ to $Z_2$ and another from $Z_2$ to $Z_1$. A single stitcher only confirms a one-way Markov blanket relation.

## 3.3. Representational Similarity

In the literature, *representational similarity* assesses whether two representations encode input information in the same way. Many studies address this question by examining the geometric structure of the representation spaces. In contrast, in this section we define representational similarity solely in terms of *what* information the representations encode. We formalize *representational similarity* using Definition 3.2.

**Definition 3.7** (Representational Similarity)**.** Let $Z_1$ and $Z_2$ be two representations, and let $X$ be the input. We say that $Z_1$ and $Z_2$ are *representationally similar* if $Z_1$ forms a Markov blanket between $Z_2$ and $X$, and $Z_2$ forms a Markov blanket between $Z_1$ and $X$. Equivalently:

$$I(X; Z_2 \mid Z_1) = I(X; Z_1 \mid Z_2) = 0. \qquad (7)$$

In other words, $Z_1$ and $Z_2$ contain exactly the same information about $X$.

### 3.4. Bridging Similarities

We now unify the notions of representational and functional similarity, demonstrating that they are not distinct phenomena but rather hierarchically related concepts governed by the complexity of the target task.

First, we establish that functional similarity is monotonic with respect to task granularity. Since processing a target variable cannot generate new information, similarity on a complex task guarantees similarity on any coarser derivative.

**Proposition 3.8** (Granular Similarity $\Rightarrow$ Coarser Similarity)**.** *Let $Z_1$ and $Z_2$ be functionally similar w.r.t. a task $Y$. Let $Y'$ be a coarser task such that $Y' = g(Y)$ for some deterministic function $g$. Then, $Z_1$ and $Z_2$ are functionally similar w.r.t. $Y'$.*

Second, we bridge the gap to representational similarity by observing that it is simply functional similarity applied to the input reconstruction task.

**Remark 3.9** (Representational as Special Case of Functional Similarity)**.** From Definitions 3.3 and 3.7, it follows that *representational similarity* is a special case of *functional similarity* where the task $Y$ is the input $X$ itself.

Finally, we unify these insights. Since any deterministic task $Y$ is derived from the input $X$ (i.e., $Y = f(X)$), $Y$ is effectively a "coarser" task relative to $X$. Therefore, Proposition 3.8 directly implies that representational similarity guarantees functional similarity for all derived tasks.

**Corollary 3.10** (Representational $\Rightarrow$ Functional Similarity)**.** *Let $X$ be an input, and let $Z_1, Z_2$ be two representations of $X$. Let $\mathcal{Y} = \{Y : Y = f(X)\}$ be the set of all deterministic tasks derived from $X$. If $Z_1$ and $Z_2$ are representationally similar, then they are functionally similar with respect to any $Y \in \mathcal{Y}$.*

Crucially, this implication is strictly one-way. While representational similarity preserves all of $X$, functional similarity only requires preserving $Y$, allowing models to differ by discarding distinct *nuisance variables* (Achille & Soatto, 2018) (task-irrelevant information).

> **Takeaway.** Functional similarity is *monotonic* with respect to task complexity.
> - *Coarse Tasks:* Similarity on a hard task guarantees similarity on any easier, coarser derivative.
> - *Representational Similarity:* As the limit case (reconstructing $X$), representational similarity is the strongest condition, guaranteeing functional similarity for *every possible deterministic task*.

## 4. *How* information is encoded

In the previous section, we defined functional and representational similarity in terms of conditional mutual information, focusing exclusively on *what* information the representations encode. However, this formulation presents three main limitations: (i) mutual information is generally difficult to compute or estimate (McAllester & Stratos, 2020), which hinders the definition of practical metrics; (ii) measuring *functional similarity* assumes access to stitchers of arbitrary complexity, which is unrealistic; and (iii) *representational similarity* definition neglects geometric aspects, deviating from most formulations in the existing literature.

To address this, we shift focus to *how* information is encoded using *usable information theory* (Section 2.1). This perspective naturally alleviates the limitations discussed above: (i) *usable* mutual information terms enable tractable computation; (ii) stitcher complexity is controlled by the predictive family $\mathcal{V}$; and (iii) geometric properties are incorporated through the transformations defined by $\mathcal{V}$. Crucially, the connection between classical and *usable* information ensures that previous statements remain valid, provided the predictive families are coherent. The main results of this section are depicted in Figure 1 and derived in Appendix A.

### 4.1. Functional Similarity under $\mathcal{V}$

Recalling Definitions 3.2 and 3.3, we say that $Z_1$ and $Z_2$ are functionally similar with respect to $Y$ if $I(Z_2; Y \mid Z_1) = I(Z_1; Y \mid Z_2) = 0$. We now extend this notion to predictive families $\mathcal{V}$.

**Definition 4.1** (Functional Similarity under $\mathcal{V}$)**.** Let $Z_1$ and $Z_2$ be two representations, $Y$ a task, and $\mathcal{V}$ a predictive family. We say that $Z_1$ and $Z_2$ are *functionally similar under $\mathcal{V}$* with respect to $Y$ if

$$I_{\mathcal{V}}(Z_2 \to Y \mid Z_1) = I_{\mathcal{V}}(Z_1 \to Y \mid Z_2) = 0. \qquad (8)$$

**How to compute functional similarity under $\mathcal{V}$.** Proposition 3.5 relates perfect stitchability to Markov blankets, establishing a formal link between stitching and functional similarity. However, infinitely expressive stitchers are rarely available in practice. We therefore move to a restricted

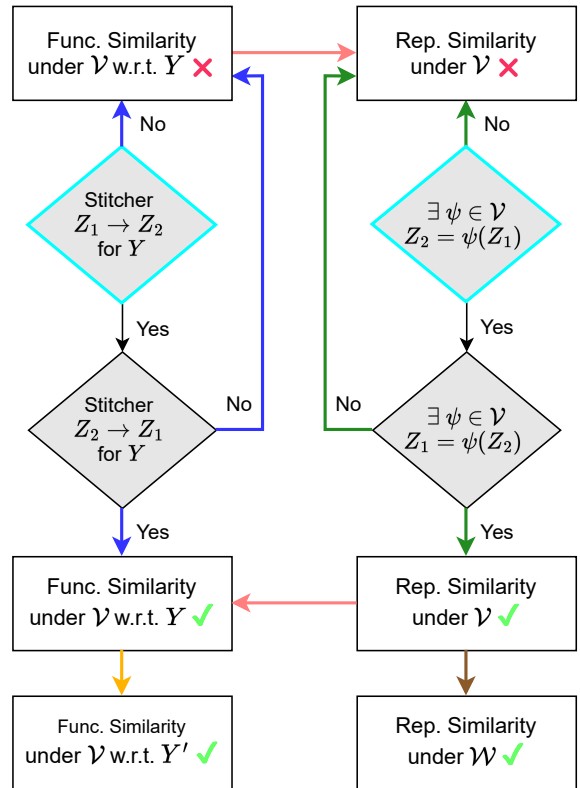

*Figure 1.* Flowchart summarizing the results of Section 4. The inputs are highlighted with cyan contours. Arrows are color-coded as follows: blue for Corollary 4.3, green for Equation 13, brown for Proposition 4.5 ($\mathcal{V} \subseteq \mathcal{W}$), amber for Proposition 4.7 ($Y' = g(Y)$), and coral for Corollary 4.8.

setting: the family of accessible stitchers $\mathcal{S}$ induces a corresponding predictive family $\mathcal{V}$, which directly connects model stitching to functional similarity under $\mathcal{V}$.

**Proposition 4.2** (Stitching as Usable Conditional Information). *Let $Z_1, Z_2$ be representations and let $h_2 : \mathcal{Z}_2 \rightarrow \mathcal{P}(\mathcal{Y})$ be a fixed task head. Let $\mathcal{S}$ be the family of accessible stitchers from $Z_1$ to $Z_2$. Defining the augmented predictive family $\mathcal{V} = \{h_2\} \cup \{h_2 \circ s : s \in \mathcal{S}\}$, we have*

$$I_{\mathcal{V}}(Z_2 \rightarrow Y \mid Z_1) = \max\left(0, \inf_{s \in \mathcal{S}} \mathbb{E}_{z_1}\Big[CE\big(y, (h_2 \circ s)[z_1](y)\big)\Big]\right.$$
$$\left. - \mathbb{E}_{z_2}\Big[CE\big(y, h_2[z_2](y)\big)\Big]\right).$$

> **Takeaway.** This result grounds model stitching in information theory: minimizing the standard Cross-Entropy loss during stitcher training is formally equivalent to minimizing the *usable conditional mutual information $I_{\mathcal{V}}(Z_2 \rightarrow Y \mid Z_1)$.* Thus, the optimal stitcher acts as a principled estimator of the usable information gap between representations.

Consequently, this allows us to adapt the equivalence established in Corollary 3.6 to the context of predictive families.

**Corollary 4.3** (Functional Similarity–Stitchability Equivalence under $\mathcal{V}$). *From Proposition 4.2, $Z_1$ and $Z_2$ are functionally similar under $\mathcal{V}$ if and only if there exist perfect stitchers $s_{12} : \mathcal{Z}_1 \rightarrow \mathcal{Z}_2$ and $s_{21} : \mathcal{Z}_2 \rightarrow \mathcal{Z}_1$ such that the composite predictors $h_2 \circ s_{12}$ and $h_1 \circ s_{21}$ belong to $\mathcal{V}$.*

To map the entropic gap in Proposition 4.2 to a practical score in $[0, 1]$, we use the ratio between stitched and native accuracy. We define the *directional* functional similarity as

$$S_{\text{func}}^{\mathcal{V}}(Z_1 \rightarrow Z_2) = \frac{\mathcal{A}(h_2 \circ s_{12})}{\mathcal{A}(h_2)}, \tag{9}$$

where $s_{12}$ is the optimal stitcher in $\mathcal{V}$. To enforce the mutual sufficiency required by Definition 4.1, we define a symmetric score as the minimum of the two directions:

$$S_{\text{func}}^{\mathcal{V}}(Z_1, Z_2) = \min\left(S_{\text{func}}^{\mathcal{V}}(Z_1 \rightarrow Z_2), \, S_{\text{func}}^{\mathcal{V}}(Z_2 \rightarrow Z_1)\right). \tag{10}$$

This score is high if and only if stitchers exist in *both* directions, and it penalizes asymmetric cases (e.g., when one representation strictly contains the other).

### 4.2. Representational Similarity under $\mathcal{V}$

Following Definitions 3.2 and 3.7, $Z_1$ and $Z_2$ are representationally similar if $I(X; Z_2 \mid Z_1) = I(X; Z_1 \mid Z_2) = 0$. We extend this notion to restricted predictive families.

**Definition 4.4** (Representational Similarity under $\mathcal{V}$). *Let $Z_1, Z_2$ be two representations and let $\mathcal{V}$ be a predictive family. We say that $Z_1$ and $Z_2$ are *representationally similar under $\mathcal{V}$* if*

$$I_{\mathcal{V}}(X \rightarrow Z_2 \mid Z_1) = I_{\mathcal{V}}(X \rightarrow Z_1 \mid Z_2) = 0. \tag{11}$$

**How to compute representational similarity under $\mathcal{V}$.** Definition 4.4 is binary, so we use a continuous score based on the usable conditional information, defined as:

$$I_{\mathcal{V}}(X \rightarrow Z_2 \mid Z_1) = H_{\mathcal{V}}(Z_2 \mid Z_1) - H_{\mathcal{V}}(Z_2 \mid X, Z_1). \tag{12}$$

To make this computation tractable, we first assume that representations are extracted via deterministic encoders ($Z_2 = f(X)$). Under this assumption, there is no intrinsic uncertainty in $Z_2$ given $X$, causing the second term, $H_{\mathcal{V}}(Z_2 \mid X, Z_1)$, to vanish.

To bridge this information-theoretic measure with standard geometric comparisons, we make a deliberate design choice: we define the predictive family $\mathcal{V}$ under a Gaussian likelihood model with fixed isotropic covariance $\sigma^2 I$. Under this formulation, minimizing the usable conditional entropy is mathematically equivalent to minimizing the Mean Squared

Error (MSE) (Bishop & Nasrabadi, 2006):

$$I_\mathcal{V}(X \to Z_2 \mid Z_1) \propto \inf_{\psi \in \mathcal{V}} \mathbb{E}\big[\|Z_2 - \psi(Z_1)\|^2\big]. \quad (13)$$

To bound this metric between 0 and 1, we normalize it by the intrinsic uncertainty $H_\mathcal{V}(Z_2 \mid \varnothing)$ (the optimal no-input predictor). Under the same Gaussian assumption, this marginal entropy is proportional to the variance, $H_\mathcal{V}(Z_2 \mid \varnothing) \propto \mathrm{Var}(Z_2)$. Subtracting this normalized ratio from 1 yields the directed representational similarity:

$$S_{\mathrm{rep}}^\mathcal{V}(Z_1 \to Z_2) = 1 - \frac{\inf_{\psi \in \mathcal{V}} \mathrm{MSE}(Z_2, \psi(Z_1))}{\mathrm{Var}(Z_2)}, \quad (14)$$

which is equivalent to the fraction of variance in $Z_2$ explained by $Z_1$ under $\mathcal{V}$ (i.e., $R^2$). Since representational similarity requires mutual reconstructibility (Definition 4.4), we define the symmetric score as the minimum of the directional components:

$$S_{\mathrm{rep}}^\mathcal{V}(Z_1, Z_2) = \min\big(S_{\mathrm{rep}}^\mathcal{V}(Z_1 \to Z_2),\ S_{\mathrm{rep}}^\mathcal{V}(Z_2 \to Z_1)\big). \quad (15)$$

This formulation enforces a strict similarity condition in Equation 11: the score is high if and only if both representations can reconstruct each other under $\mathcal{V}$, preventing high similarity in cases of asymmetric information containment.

> **Takeaway.** Reconstruction-based metrics (e.g., Procrustes alignment or linear $L_2$ regression) can be interpreted as estimators of $I_\mathcal{V}(X \to Z_2 \mid Z_1)$. Normalizing reconstruction error by $\mathrm{Var}(Z_1)$ yields a bounded similarity score equivalent to $R^2$.

**Proposition 4.5** (Monotonicity of Representational Similarity). *Let $Z_1, Z_2$ be representations and let $\mathcal{V} \subseteq \mathcal{W}$ be predictive families. Then*

$$S_{rep}^\mathcal{V}(Z_1, Z_2) \le S_{rep}^\mathcal{W}(Z_1, Z_2). \quad (16)$$

This highlights that similarity is relative to the observer's capacity: representations can appear distinct under a restrictive $\mathcal{V}$ (e.g., linear maps) yet equivalent under a richer $\mathcal{W}$ (e.g., non-linear networks).

**Hypothesis 4.6** (Alignment of Metrics via Predictive Families). *Let $M$ be a similarity metric (e.g., CKA, SVCCA) invariant to a transformation group $\mathcal{G}$. Let $\mathcal{V}$ be the predictive family associated with $\mathcal{G}$ (as formalized by Williams et al., 2021). We hypothesize*

$$\mathrm{Corr}\Big(M(Z_1, Z_2),\ S_{rep}^\mathcal{V}(Z_1, Z_2)\Big) \gg 0. \quad (17)$$

**Intuition.** $M$ assesses alignment modulo a transformation group $\mathcal{G}$. Since $S_{\mathrm{rep}}^\mathcal{V}$ measures reconstructibility using the

corresponding family $\mathcal{V}$, both metrics penalize the same structural differences—namely those that cannot be bridged by the allowed transformations (Almudévar et al., 2025).

If this hypothesis holds, it effectively unifies standard metrics as estimators of *usable information*, each characterized by its own predictive family $\mathcal{V}$.

### 4.3. Bridging Similarities under $\mathcal{V}$

We now extend our hierarchy to the realistic setting of usable information. By restricting our analysis to a predictive family $\mathcal{V}$, we demonstrate that the relationship established in Section 3.4 holds even under computational constraints.

First, we establish that functional similarity under $\mathcal{V}$ remains monotonic with respect to task granularity.

**Proposition 4.7** (Granular Similarity $\Rightarrow$ Coarser Similarity under $\mathcal{V}$). *Let $Z_1$ and $Z_2$ be functionally similar under predictive family $\mathcal{V}$ with respect to task $Y$. Let $Y' = g(Y)$ be a coarser task. If the family $\mathcal{V}$ is closed under post-composition with $g$ (i.e., $\forall h \in \mathcal{V}$, $g \circ h \in \mathcal{V}$), then $Z_1$ and $Z_2$ are functionally similar under $\mathcal{V}$ with respect to $Y'$.*

Intuitively, if a stitcher aligns two representations well enough to solve a complex task, that alignment will also suffice for any simpler, derived task. Since the coarsening occurs at the output level, the original stitcher remains valid and can be reused without modification. We now derive the main bridging result.

**Corollary 4.8** (Representational $\Rightarrow$ Functional Similarity under $\mathcal{V}$). *Let $\mathcal{V}$ be a predictive family that is closed under composition. If $Z_1$ and $Z_2$ are representationally similar under $\mathcal{V}$ (losslessly mappable via $\mathcal{V}$), then they are functionally similar under $\mathcal{V}$ with respect to any deterministic task $Y$ derived from $X$.*

## 5. Experiments

In this section, we evaluate 58 trained models (spanning 14 distinct architectures) across multiple datasets, and compare every intra-dataset model pair with each other to yield 12,459 distinct layer-to-layer comparisons. Full architecture, dataset, and extraction details are in Appendix C. The goal of these experiments is not to analyze the specific similarities for any given pair of layers, but rather to empirically validate the properties and theoretical connections described in the previous sections through a large-scale analysis.

**Datasets** We use MNIST (LeCun et al., 2002), CIFAR-10, CIFAR-100 (Krizhevsky et al., 2009), SVHN (Goodfellow et al., 2013), Tiny-Imagenet (Le & Yang, 2015), and ImageNet (Deng et al., 2009). These span from grayscale digits to natural images with coarse and fine-grained labels.

**Encoder Architectures** To ensure robust generalization

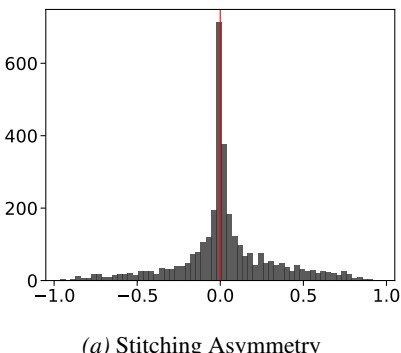

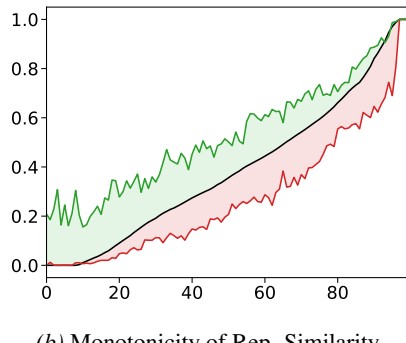

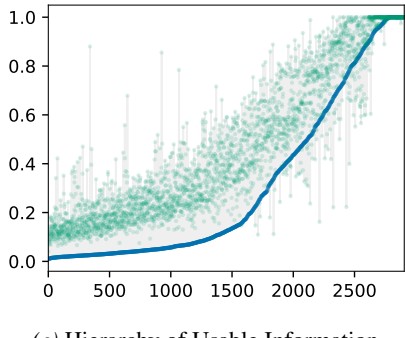

*(a)* Stitching Asymmetry  *(b)* Monotonicity of Rep. Similarity  *(c)* Hierarchy of Usable Information

*Figure 2.* **Experimental Validation.** (a) The wide distribution of accuracy differences (x-axis) confirms that stitching is often asymmetric. (b) Representational similarity (y-axis) across layer pairs sorted by their *Ortho+Scale* similarity. The similarity hierarchy (Affine > Ortho+Scale > Ortho) confirms that representational similarity increases as the predictive family $\mathcal{V}$ becomes more expressive. (c) Functional similarity (y-axis) across layer pairs sorted by their fine-task similarity. Coarse-task similarity (green) consistently upper-bounds fine-task similarity (blue), validating that usable information is strictly hierarchical with respect to task granularity.

across model types and scales, we evaluate a diverse pool of encoders. For the smaller-scale datasets, this includes five linear architectures, two simple convolutional networks, and adaptations of modern deep architectures for $32 \times 32$ and $64 \times 64$ inputs: ResNet20 (He et al., 2016), DenseNet40 (Huang et al., 2017), ShuffleNetV2 (Ma et al., 2018), and MobileNetV3 (Howard et al., 2019) (detailed in Appendix B). For the ImageNet dataset, we utilize standard pretrained ResNet-18, ResNet-34, and ResNet-50 models.

**Stitchers** To isolate representational differences from structural incompatibilities, we strictly filter out cross-architecture stitching, restricting all pairwise comparisons to be exclusively intra-architecture (CNN-to-CNN and MLP-to-MLP). For these within-architecture pairs, we employ linear layers as stitchers for MLP encoders and $1 \times 1$ convolutions for CNNs. We analyze three specific families of these stitchers: **affine** (unconstrained), **(quasi-)orthogonal with isotropic scaling**, and **(quasi-)orthogonal**. The affine variant minimizes classification loss, whereas the orthogonal types incorporate regularization terms to enforce geometric constraints (Bansal et al., 2018), as detailed in Appendix D.

**Similarity Metrics** For **functional similarity**, we use the accuracy ratio (Eq. 9). For **representational similarity**, we employ standard metrics (RSA, CKA, SVCCA) alongside the normalized negative MSE (Eq. 14) for affine, (quasi-)orthogonal+isotropic scaling, and (quasi-)orthogonal families. These are obtained by minimizing the MSE—directly for affine, and with regularization for the latter two.

### 5.1. Stitching Asymmetry

To validate Corollary 3.6, we assessed the symmetry of stitching performance. For every representation pair $(Z_A, Z_B)$, we trained forward ($s_{A \to B}$) and reverse ($s_{B \to A}$) stitchers, calculating the difference in their *accuracy ratios*.

The results, shown in Figure 2a, reveal a distribution with a peak at zero—likely driven by *layer proximity* where adjacent representations remain nearly identical (Kornblith et al., 2019). However, the distribution also exhibits heavy tails with significant variance. This confirms that stitching is frequently asymmetric: $Z_1$ often predicts $Z_2$ (forming a Markov blanket), while $Z_2$ fails to reconstruct $Z_1$ due to information loss. This empirical evidence supports our takeaway: functional similarity cannot be established by a single stitcher; it requires bidirectional validation.

### 5.2. Monotonicity of Representational Similarity

To validate the *Monotonicity of Representational Similarity* (Proposition 4.5), we examined how the choice of predictive family $\mathcal{V}$ alters the representational similarity. We computed the transformations that minimize the Mean Squared Error (MSE) between all representation pairs (as per Eq. 13) under three nested families: *Orthogonal* (equivalent to Procrustes Analysis), *Orthogonal + Isotropic Scaling*, and *Affine* (standard least-squares).

Figure 2b displays the resulting representational similarity (measured as Eq. 14) for these pairs, sorted along the x-axis by their *Orthogonal+Scaling* similarity. To ensure visual clarity across the thousands of layer pairs, the curves are summarized into 100 equal-count bins by averaging within bins; however, we verified that the strict inequality holds for *all* individual samples. The plot confirms the theoretical hierarchy: the *Affine* transformations (green) yield the highest similarity, followed by *Orthogonal + Scaling* (black), and finally *Orthogonal* (red). This illustrates that "similarity" is relative: representations that appear distant under rigid Procrustes constraints may appear nearly identical when viewed through the more expressive Affine lens.

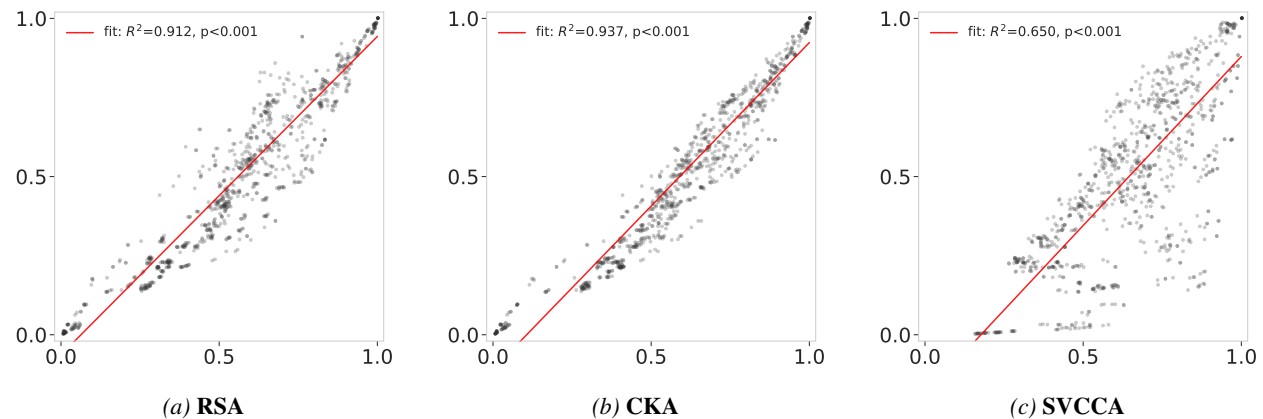

*(a)* **RSA**                *(b)* **CKA**                *(c)* **SVCCA**

*Figure 3.* **Metric Alignment.** Comparison between standard similarity metrics (y-axis) and the representational similarity derived from an Orthogonal + Scaling stitcher (x-axis). The strong correlations (especially for CKA and RSA) support the hypothesis that these metrics effectively measure usable information under specific transformation groups.

### 5.3. Standard Metrics as Proxies for Usable Information

To validate the *Alignment of Metrics via Predictive Families* (Hypothesis 4.6), we compared standard similarity metrics (RSA, CKA, SVCCA) against our proposed representational similarity measure (Eq. 14). We evaluated these using the *Orthogonal + Isotropic Scaling* stitcher, as linear CKA and RSA are theoretically invariant to unitary transformations of activation spaces. Crucially, this analysis is restricted to MLP architectures; applying spatially agnostic metrics like CKA and RSA to convolutional networks introduces mathematical inconsistency when evaluated against the spatially aware $1 \times 1$ convolutional stitchers in our framework.

As shown in Figure 3, we observe strong alignment for CKA ($R^2 = 0.937$) and RSA ($R^2 = 0.912$), suggesting these metrics effectively serve as proxies for usable information under orthogonal constraints. In contrast, SVCCA demonstrates a moderate correlation ($R^2 = 0.650$). This divergence likely stems from the hard thresholding inherent in SVCCA, which retains only the top-$k$ singular vectors; this implicitly assumes a low-rank predictive family, thereby discarding distributed information that the stitcher successfully exploits. These results support a unified framework where widely used metrics are interpreted as estimators of usable information, each characterized by a specific implicit predictive family.

### 5.4. Hierarchy of Functional Similarity

In order to validate the hierarchical monotonicity of functional similarity (Proposition 4.7), we utilized the CIFAR-100 dataset, which inherently possesses a canonical two-level hierarchy: 100 *fine* classes and 20 *coarse* superclasses. This allows us to define two distinct tasks, $Y$ (fine) and $Y'$ (coarse).

Crucially, we employed the exact same set of encoders—trained originally on the standard 100-class task—for both evaluations. Consequently, the representations $Z_1$ and $Z_2$ remain completely identical across both experimental settings; only the target task changes. To establish the native baseline performance for the coarse task, we froze these pre-trained encoders and trained a new linear task head targeting the 20 superclasses.

We computed the functional similarity for every representation pair using all three stitcher families. Figure 2c displays the results, sorting pairs along the x-axis by their functional similarity on the fine task. The data reveals a strict dominance relationship: the functional similarity on the coarse task (green) consistently acts as an upper bound to the similarity on the fine task (blue). This confirms that representations frequently diverge by discarding fine-grained nuisances while simultaneously preserving the high-level semantic structure required for coarser tasks. We attribute rare violations in the high-similarity regime to optimization noise, as vanishing error margins near saturation make the ratio metric sensitive to minor numerical fluctuations.

### 5.5. Sufficiency of Representational Similarity

To empirically validate Corollary 4.8—which posits that representational similarity implies functional similarity—we analyzed the relationship between these two measures under three predictive families: *Orthogonal*, *Orthogonal + Scaling*, and *Affine*.

Figure 4 plots the probability of observing high functional similarity (accuracy ratio > 0.95) as a function of representational similarity (Eq. 14). Across all three families, we observe a consistent trend: as representational similarity approaches 1.0, the probability of functional similarity converges to 1.0. This confirms that representational similarity is a *sufficient* condition for functional similarity.

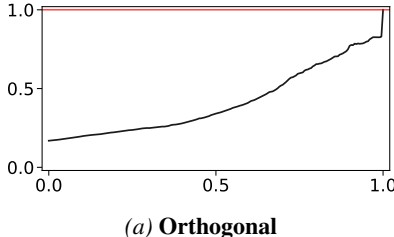
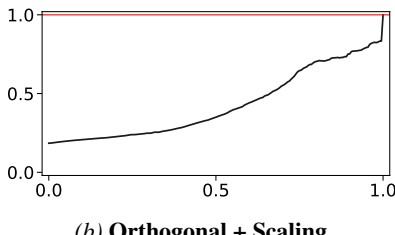
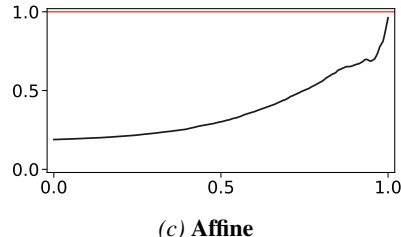

*(a)* **Orthogonal**      *(b)* **Orthogonal + Scaling**      *(c)* **Affine**

*Figure 4.* **Rep. Similarity Implies Func. Similarity.** Conditional probability of high functional similarity (y-axis) given representational similarity (x-axis); the red horizontal lines represent the theoretical maximum at $y = 1$. While high representational similarity guarantees functional equivalence (validating Corollary 4.8), the reverse does not hold: high functional similarity frequently occurs despite low representational similarity.

However, the curves also reveal that high functional similarity frequently occurs even when representational similarity is low. This asymmetry highlights that representational similarity is not a *necessary* condition. Intuitively, this occurs because functional similarity requires only the shared encoding of task-relevant information ($Y$), whereas representational similarity demands the alignment of all encoded information (including task-irrelevant features of $X$). Thus, two representations can solve a task identically while remaining geometrically distinct.

## 6. Discussion and Limitations

While our framework unifies functional and representational similarity, the formal proofs rely on strict theoretical constraints, including deterministic encoders and closure properties of the transformation family $\mathcal{V}$. Separately, our formulation rejects the notion of an off-the-shelf, universal similarity metric. Because similarity is inherently tied to usable information, our framework requires practitioners to explicitly select or approximate $\mathcal{V}$. This necessity is not a disadvantage, but rather a crucial design freedom, allowing practitioners to dictate what "similar" means based on the specific capacity and constraints of their target application.

Translating this framework into practice introduces distinct empirical challenges. Estimating usable information requires training intermediate stitchers, making the resulting scores sensitive to optimization noise, threshold choices, and stitcher capacity. While standard representational metrics (CKA, RSA, SVCCA) bypass these optimization hurdles, our formulation establishes them merely as empirical proxies rather than strict theoretical estimators. For mathematical consistency, we restrict our analysis of these proxies to MLP architectures, avoiding the application of spatially agnostic measures to convolutional features.

Empirically, our validation is restricted to supervised vision models, and all pairwise similarity calculations are strictly intra-architecture (MLP-to-MLP and Conv-to-Conv). We do not explore attention-based models like Transformers; evaluating how these theoretical relationships transfer to

such architectures or other domains (e.g., NLP) remains an open question.

## 7. Conclusion

In this work, we introduced a unified framework for analyzing neural representations through the lens of *usable information*. Our theoretical proofs and large-scale empirical analysis yield insights across three key dimensions:

**Functional Similarity** We established a formal link between the stitching loss function and usable conditional mutual information. Furthermore, our experiments revealed that stitching is frequently asymmetric, confirming that a robust measure of functional similarity requires a bidirectional analysis rather than a single unidirectional stitcher.

**Representational Similarity** We proved that reconstruction-based metrics (such as L2 or Procrustes) have a clear interpretation in terms of usable conditional mutual information. Empirically, we showed that classical metrics (CKA, RSA, SVCCA) strongly correlate with these measures, validating them as estimators of usable information under specific constraints. Crucially, we demonstrated that representational similarity is strictly dependent on the capacity of the predictive family—what appears distinct under a rigid transformation may be identical under a more expressive one.

**Connection Between Similarities** We validated that representational similarity is a *sufficient* but not *necessary* condition for functional similarity. We unify these concepts via a task-granularity hierarchy: alignment on complex tasks guarantees alignment on coarser derivatives, identifying representational similarity as the limit of maximum granularity: input reconstruction.

## Acknowledgements

This work has received funding from MCIN/AEI/10.13039/501100011033 under Grant PID2024-155948OB-C53.

## Impact Statement

This paper presents work whose goal is to advance the field of Machine Learning. There are many potential societal consequences of our work, none which we feel must be specifically highlighted here.

## Code Availability

An implementation accompanying this work is available at https://github.com/antonioalmudevar/bridging_similarities

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

# A. Mathematical Proofs

## A.1. Proof of Proposition 3.5

**Proposition 3.5** (Markov Blanket–Stitchability Equivalence)**.** Let $q_{\phi_2}$ be a Bayes-optimal predictor for $Z_2$. Then, $Z_1$ is a Markov blanket for $Y$ relative to $Z_2$ if and only if $Z_1$ is perfectly stitchable into $Z_2$.

*Proof.* Let $H(Y \mid Z)$ denote the conditional entropy, which constitutes the theoretical lower bound of the Cross Entropy (CE) for any predictor using representation $Z$.

($\Rightarrow$) **Markov Blanket implies Stitchability.** Assume $Z_1$ is a Markov blanket for $Y$ relative to $Z_2$. By definition, this means $I(Y; Z_2 \mid Z_1) = 0$, which mathematically implies that $Z_1$ contains all the information about $Y$ that is present in $Z_2$. Formally, this establishes:

$$H(Y \mid Z_1) \leq H(Y \mid Z_2)$$

By assumption, $q_{\phi_2}$ is Bayes-optimal for $Z_2$, meaning its loss achieves the exact information-theoretic limit: $CE(q_{\phi_2}(Z_2)) = H(Y \mid Z_2)$. To proceed, we explicitly assume that the stitcher family $\mathcal{S}$ is a universal approximator class. Under this assumption, there exists a stitcher $s \in \mathcal{S}$ capable of preserving and extracting the task-relevant information from $Z_1$ such that the composed predictor $q_{\phi_2} \circ s$ optimally leverages $Z_1$. Therefore:

$$CE(q_{\phi_2} \circ s(Z_1)) = H(Y \mid Z_1) \leq H(Y \mid Z_2) = CE(q_{\phi_2}(Z_2))$$

This directly satisfies the condition for perfect stitchability.

($\Leftarrow$) **Stitchability implies Markov Blanket.** Assume $Z_1$ is perfectly stitchable into $Z_2$. By definition, there exists a stitcher $s \in \mathcal{S}$ such that the composed predictor evaluates at least as well as the native head:

$$CE(q_{\phi_2} \circ s(Z_1)) \leq CE(q_{\phi_2}(Z_2))$$

Because $q_{\phi_2}$ is Bayes-optimal for $Z_2$, the right-hand side is identically $H(Y \mid Z_2)$. On the left-hand side, the composite predictor $q_{\phi_2} \circ s$ is a function acting exclusively on $Z_1$. By fundamental information theory bounds, the CE of any predictor acting strictly on $Z_1$ is lower-bounded by $H(Y \mid Z_1)$. We therefore obtain the following chain of inequalities:

$$H(Y \mid Z_1) \leq CE(q_{\phi_2} \circ s(Z_1)) \leq CE(q_{\phi_2}(Z_2)) = H(Y \mid Z_2)$$

The resulting inequality $H(Y \mid Z_1) \leq H(Y \mid Z_2)$ establishes that $Z_2$ yields no additional task-relevant information about $Y$ beyond what is inherently available in $Z_1$. Consequently, $I(Y; Z_2 \mid Z_1) = 0$, fulfilling the formal definition of a Markov blanket. $\square$

## A.2. Proof of Corollary 3.6

**Corollary 3.6** (Functional Similarity–Stitchability Equivalence)**.** $Z_1$ and $Z_2$ are functionally similar w.r.t. $Y$ if and only if $Z_1$ is perfectly stitchable into $Z_2$ and vice versa.

*Proof.* The proof follows directly from the definition of Functional Similarity (Definition 3.3) and the application of Proposition 3.5 in both directions.

($\Rightarrow$) **Functional Similarity implies Mutual Stitchability:**
Assume $Z_1$ and $Z_2$ are functionally similar w.r.t. $Y$. By Definition 3.3, this entails two conditions:

  (i) $Z_1$ forms a Markov blanket between $Z_2$ and $Y$ (i.e., $I(Y; Z_2 \mid Z_1) = 0$).

 (ii) $Z_2$ forms a Markov blanket between $Z_1$ and $Y$ (i.e., $I(Y; Z_1 \mid Z_2) = 0$).

Applying Proposition 3.5 to condition (i), since $Z_1$ is a Markov blanket for $Z_2$, $Z_1$ is perfectly stitchable into $Z_2$. Applying Proposition 3.5 symmetrically to condition (ii), since $Z_2$ is a Markov blanket for $Z_1$, $Z_2$ is perfectly stitchable into $Z_1$. Thus, functional similarity implies mutual stitchability.

($\Leftarrow$) **Mutual Stitchability implies Functional Similarity:**
Assume $Z_1$ is perfectly stitchable into $Z_2$ and $Z_2$ is perfectly stitchable into $Z_1$. From Proposition 3.5, the stitchability of

$Z_1$ into $Z_2$ implies that $Z_1$ forms a Markov blanket between $Z_2$ and $Y$. Similarly, the stitchability of $Z_2$ into $Z_1$ implies that $Z_2$ forms a Markov blanket between $Z_1$ and $Y$.

Since both Markov blanket conditions hold simultaneously:

$$I(Y; Z_2 \mid Z_1) = 0 \quad \text{and} \quad I(Y; Z_1 \mid Z_2) = 0 \tag{18}$$

This satisfies Definition 3.3, proving that $Z_1$ and $Z_2$ are functionally similar. $\square$

### A.3. Proof of Proposition 3.8

**Proposition 3.8** (Granular Similarity $\Rightarrow$ Coarser Similarity)**.** Let $Z_1$ and $Z_2$ be functionally similar w.r.t. a task $Y$. Let $Y'$ be a *coarser* task such that $Y' = g(Y)$ for some deterministic function $g$. Then, $Z_1$ and $Z_2$ are functionally similar w.r.t. $Y'$.

*Proof.* By Definition 3.3, functional similarity with respect to task $Y$ requires that $Z_1$ and $Z_2$ are perfectly stitchable into each other. Following our established equivalence, this mutual stitchability means $Z_1$ and $Z_2$ act as Markov blankets for one another relative to $Y$. This formally establishes the dual Markov chains $Y \leftrightarrow Z_1 \leftrightarrow Z_2$ and $Y \leftrightarrow Z_2 \leftrightarrow Z_1$.

Consequently, the representations share the exact same task-relevant information, which requires their conditional distributions to be identical:

$$P(Y \mid Z_1) = P(Y \mid Z_1, Z_2) = P(Y \mid Z_2). \tag{19}$$

Now, let $Y' = g(Y)$ be a coarser task. The predictive distribution for this coarser task is obtained by marginalizing the fine-grained distribution over the preimage of $g$:

$$P(Y' = y' \mid Z) = \sum_{y \in g^{-1}(y')} P(Y = y \mid Z). \tag{20}$$

Substituting the equivalence of the fine-grained conditional distributions into this summation, we obtain:

$$P(Y' = y' \mid Z_1) = \sum_{y \in g^{-1}(y')} P(Y = y \mid Z_1) = \sum_{y \in g^{-1}(y')} P(Y = y \mid Z_2) = P(Y' = y' \mid Z_2). \tag{21}$$

Since the predictive distributions for $Y'$ are strictly identical, $Z_1$ and $Z_2$ achieve the same Bayes-optimal risk for $Y'$, directly satisfying the definition of functional similarity for the coarser task. $\square$

### A.4. Proof of Corollary 3.10

**Corollary 3.10** (Representational $\Rightarrow$ Functional Similarity)**.** Let $X$ be an input, and let $Z_1, Z_2$ be two representations of $X$. Let $\mathcal{Y} = \{Y : Y = f(X)\}$ be the set of all deterministic tasks derived from $X$. If $Z_1$ and $Z_2$ are representationally similar, then they are functionally similar with respect to any $Y \in \mathcal{Y}$.

*Proof.* By Definition 3.7, representational similarity requires that $I(X; Z_2 \mid Z_1) = I(X; Z_1 \mid Z_2) = 0$.

In information theory, $I(X; Z_2 \mid Z_1) = 0$ is the exact mathematical equivalent of conditional independence: $X \perp\!\!\!\perp Z_2 \mid Z_1$. This establishes the Markov chain $Z_2 \leftrightarrow Z_1 \leftrightarrow X$.

Let $Y \in \mathcal{Y}$ be any task deterministically derived from $X$, such that $Y = f(X)$. A fundamental property of conditional independence (and the Data Processing Inequality) states that if a variable is conditionally independent of another, any deterministic function applied to it remains conditionally independent. Because $Y$ is strictly a downstream function of $X$, the Markov chain naturally extends to $Z_2 \leftrightarrow Z_1 \leftrightarrow X \leftrightarrow Y$.

Consequently, $Y$ must also be conditionally independent of $Z_2$ given $Z_1$ ($Y \perp\!\!\!\perp Z_2 \mid Z_1$), which identically evaluates to:

$$I(Y; Z_2 \mid Z_1) = 0. \tag{22}$$

By applying the exact same symmetric logic to the reverse direction ($Z_1 \rightarrow Z_2$), we obtain $I(Y; Z_1 \mid Z_2) = 0$. Because the conditional mutual information between the representations and the task is strictly zero in both directions, $Z_1$ and $Z_2$ are functionally similar for any deterministic task $Y \in \mathcal{Y}$, natively accommodating both discrete and continuous domains. $\square$

### A.5. Proof of Proposition 4.2

**Proposition 4.2** (Stitching as Usable Conditional Information). Let $Z_1, Z_2$ be representations and let $h_2 : \mathcal{Z}_2 \to \mathcal{P}(\mathcal{Y})$ be a fixed task head. Let $\mathcal{S}$ be the family of accessible stitchers from $Z_1$ to $Z_2$. Defining the augmented predictive family $\mathcal{V} = \{h_2\} \cup \{h_2 \circ s : s \in \mathcal{S}\}$, we have

$$I_\mathcal{V}(Z_2 \to Y \mid Z_1) = \max \left( 0, \inf_{s \in \mathcal{S}} \mathbb{E}_{z_1} \left[ CE(y, (h_2 \circ s)[z_1](y)) \right] - \mathbb{E}_{z_2} \left[ CE(y, h_2[z_2](y)) \right] \right).$$

*Proof.* We use the definition of Conditional Usable Information from Xu et al. (2020). For a predictive family $\mathcal{V}$, the conditional usable information is:

$$I_\mathcal{V}(Z_2 \to Y \mid Z_1) = H_\mathcal{V}(Y \mid Z_1) - H_\mathcal{V}(Y \mid Z_1, Z_2). \tag{23}$$

The family $\mathcal{V}$ is defined as the union of the original head and the stitched heads: $\mathcal{V} = \{h_2\} \cup \{h_2 \circ s : s \in \mathcal{S}\}$. We evaluate the two terms in Eq. (23):

**1. The Conditional Entropy** $H_\mathcal{V}(Y \mid Z_1)$**:** Given only $Z_1$, we are restricted to the subset of $\mathcal{V}$ that operates on $Z_1$. This corresponds to the stitched predictors:

$$H_\mathcal{V}(Y \mid Z_1) = \inf_{s \in \mathcal{S}} \mathbb{E}_{z_1} \left[ CE(y, (h_2 \circ s)[z_1](y)) \right]. \tag{24}$$

**2. The Joint Entropy** $H_\mathcal{V}(Y \mid Z_1, Z_2)$**:** This term represents the minimum risk achievable by selecting a function $f \in \mathcal{V}$ given access to both $Z_1$ and $Z_2$. Since functions in $\mathcal{V}$ are restricted to take either $Z_2$ (the head $h_2$) or $Z_1$ (the stitched heads) as input, the joint risk is the minimum of the risks of the individual components. Substituting Eq. (24), we obtain:

$$\begin{aligned} H_\mathcal{V}(Y \mid Z_1, Z_2) &= \min \left( \mathbb{E}_{z_2} \left[ CE(y, h_2[z_2](y)) \right], \inf_{s \in \mathcal{S}} \mathbb{E}_{z_1} \left[ CE(y, (h_2 \circ s)[z_1](y)) \right] \right) \\ &= \min \left( \mathbb{E}_{z_2} \left[ CE(y, h_2[z_2](y)) \right], H_\mathcal{V}(Y \mid Z_1) \right). \end{aligned} \tag{25}$$

**3. Final Substitution:** Substituting Eq. (25) back into the definition of usable information (Eq. (23)), we get:

$$I_\mathcal{V}(Z_2 \to Y \mid Z_1) = H_\mathcal{V}(Y \mid Z_1) - \min \left( \mathbb{E}_{z_2} \left[ CE(y, h_2[z_2](y)) \right], H_\mathcal{V}(Y \mid Z_1) \right). \tag{26}$$

Using the algebraic identity $A - \min(B, A) = \max(A - B, 0)$, this simplifies exactly to:

$$\begin{aligned} I_\mathcal{V}(Z_2 \to Y \mid Z_1) &= \max \left( 0, H_\mathcal{V}(Y \mid Z_1) - \mathbb{E}_{z_2} \left[ CE(y, h_2[z_2](y)) \right] \right) \\ &= \max \left( 0, \inf_{s \in \mathcal{S}} \mathbb{E}_{z_1} \left[ CE(y, (h_2 \circ s)[z_1](y)) \right] - \mathbb{E}_{z_2} \left[ CE(y, h_2[z_2](y)) \right] \right). \end{aligned}$$

$\square$

### A.6. Proof of Corollary 4.3

**Corollary 4.3** (Functional Similarity–Stitchability Equivalence under $\mathcal{V}$). From Proposition 4.2, $Z_1$ and $Z_2$ are functionally similar under $\mathcal{V}$ if and only if there exist perfect stitchers $s_{12} : \mathcal{Z}_1 \to \mathcal{Z}_2$ and $s_{21} : \mathcal{Z}_2 \to \mathcal{Z}_1$ such that the composite predictors $h_2 \circ s_{12}$ and $h_1 \circ s_{21}$ belong to $\mathcal{V}$.

*Proof.* The definition of Functional Similarity under $\mathcal{V}$ (Definition 4.1) requires:

$$I_\mathcal{V}(Z_2 \to Y \mid Z_1) = 0 \quad \text{and} \quad I_\mathcal{V}(Z_1 \mid Z_2 \to Y) = 0. \tag{27}$$

Using the result from Proposition 4.2, we expand the first term:

$$I_\mathcal{V}(Z_2 \to Y \mid Z_1) = \inf_{s \in \mathcal{S}_{1 \to 2}} \mathbb{E}[CE(h_2 \circ s)] - \mathbb{E}[CE(h_2)]. \tag{28}$$

For this term to be zero, the infimum of the stitched risk must exactly equal the risk of the original head $h_2$. This implies there exists a stitcher $s_{12} \in \mathcal{S}_{1 \to 2}$ (or a sequence of stitchers approaching the limit) such that:

$$\mathbb{E}_{z_1}[CE(y, (h_2 \circ s_{12})(z_1))] = \mathbb{E}_{z_2}[CE(y, h_2(z_2))]. \tag{29}$$

This is the definition of a "perfect stitcher" within the constrained family $\mathcal{V}$ (i.e., the stitched model performs indistinguishably from the native model).

By symmetry, for the second term $I_\mathcal{V}(Z_1 \mid Z_2 \to Y) = 0$, there must exist a reverse stitcher $s_{21} \in \mathcal{S}_{2 \to 1}$ such that the stitched predictor $h_1 \circ s_{21}$ achieves the same risk as the native predictor $h_1$.

Therefore, $I_\mathcal{V}(Z_2 \to Y \mid Z_1) = I_\mathcal{V}(Z_1 \mid Z_2 \to Y) = 0$ is equivalent to the existence of perfect stitchers in both directions. $\qquad\square$

### A.7. Derivation of Equation 14

In this section, we formally justify using Mean Squared Error (MSE) as a proxy for the conditional mutual information $I_\mathcal{V}(X \to Z_2 \mid Z_1)$.

*Proof.* Recall the definition of Conditional Mutual Information:

$$I(X; Z_2 \mid Z_1) = H(Z_2 \mid Z_1) - H(Z_2 \mid X, Z_1). \tag{30}$$

In the context of standard deep learning, the representation $Z_2$ is a deterministic function of the input $X$ (i.e., $Z_2 = f(X)$). Consequently, given $X$, there is no uncertainty in $Z_2$. Thus, the second term vanishes:[2]

$$H(Z_2 \mid X, Z_1) = 0. \tag{31}$$

This simplifies the objective to minimizing the conditional entropy (or strictly, the cross-entropy) of $Z_2$ given $Z_1$ under the predictive family $\mathcal{V}$:

$$I_\mathcal{V}(X \to Z_2 \mid Z_1) = H_\mathcal{V}(Z_2 \mid Z_1) \equiv \inf_{\psi \in \mathcal{V}} \mathbb{E}_{p(z_1, z_2)}[-\log q_\psi(z_2 \mid z_1)], \tag{32}$$

where $q_\psi$ is the predictive distribution parameterized by $\psi$. We assume a Gaussian likelihood model with fixed variance $\sigma^2$, which is standard for regression tasks (Bishop & Nasrabadi, 2006):

$$q_\psi(z_2 \mid z_1) = \mathcal{N}(z_2 \mid \psi(z_1), \sigma^2 I). \tag{33}$$

The negative log-likelihood for a Gaussian is:

$$-\log q_\psi(z_2 \mid z_1) = \frac{1}{2\sigma^2} \|z_2 - \psi(z_1)\|^2 + \frac{d}{2} \log(2\pi\sigma^2). \tag{34}$$

Since $\sigma^2$ is fixed, minimizing the usable information $I_\mathcal{V}$ is equivalent to minimizing the expected squared error:

$$\operatorname*{argmin}_{\psi \in \mathcal{V}} I_\mathcal{V}(X \to Z_2 \mid Z_1) \equiv \operatorname*{argmin}_{\psi \in \mathcal{V}} \mathbb{E}\left[\|z_2 - \psi(z_1)\|^2\right]. \tag{35}$$

Thus, we write the proportionality in the optimization sense:

$$I_\mathcal{V}(X \to Z_2 \mid Z_1) \propto \inf_{\psi \in \mathcal{V}} \mathrm{MSE}(Z_2, \psi(Z_1)). \tag{36}$$

$\qquad\square$

---

[2]In cases of stochastic representations (e.g., VAEs or networks with dropout), this term is a constant determined by the encoder's variance and does not depend on the relationship between $Z_1$ and $Z_2$.

## A.8. Proof of Proposition 4.5

**Proposition 4.5** (Monotonicity of Representational Similarity). Let $Z_1, Z_2$ be representations and let $\mathcal{V} \subseteq \mathcal{W}$ be predictive families. Then

$$S_{\text{rep}}^{\mathcal{V}}(Z_1, Z_2) \leq S_{\text{rep}}^{\mathcal{W}}(Z_1, Z_2).$$

*Proof.* Let $S_{\text{rep}}^{\mathcal{V}}(Z_2 \to Z_1)$ denote the directed representational similarity under the predictive family $\mathcal{V}$. By definition (Equation 14), we have:

$$S_{\text{rep}}^{\mathcal{V}}(Z_2 \to Z_1) = 1 - \frac{\inf_{\psi \in \mathcal{V}} \mathbb{E}[\|Z_1 - \psi(Z_2)\|^2]}{\text{Var}(Z_1)}. \tag{37}$$

Since $\mathcal{V} \subseteq \mathcal{W}$, the set of functions over which we minimize the Mean Squared Error (MSE) in $\mathcal{W}$ includes all functions in $\mathcal{V}$. Therefore, the minimum error achievable in $\mathcal{W}$ cannot be greater than that in $\mathcal{V}$:

$$\inf_{\phi \in \mathcal{W}} \mathbb{E}[\|Z_1 - \phi(Z_2)\|^2] \leq \inf_{\psi \in \mathcal{V}} \mathbb{E}[\|Z_1 - \psi(Z_2)\|^2]. \tag{38}$$

Since the variance $\text{Var}(Z_1)$ is strictly positive and independent of the predictive family, substituting Inequality (38) into the similarity definition yields:

$$1 - \frac{\inf_{\phi \in \mathcal{W}} \mathbb{E}[\|Z_1 - \phi(Z_2)\|^2]}{\text{Var}(Z_1)} \geq 1 - \frac{\inf_{\psi \in \mathcal{V}} \mathbb{E}[\|Z_1 - \psi(Z_2)\|^2]}{\text{Var}(Z_1)} \tag{39}$$

$$\implies S_{\text{rep}}^{\mathcal{W}}(Z_2 \to Z_1) \geq S_{\text{rep}}^{\mathcal{V}}(Z_2 \to Z_1). \tag{40}$$

This monotonicity holds for both directions ($Z_2 \to Z_1$ and $Z_1 \to Z_2$). Finally, let $S_{\text{rep}}^{\mathcal{V}}(Z_1, Z_2) = \min(a, b)$ and $S_{\text{rep}}^{\mathcal{W}}(Z_1, Z_2) = \min(A, B)$, where $a, b$ are the directed scores under $\mathcal{V}$ and $A, B$ are the directed scores under $\mathcal{W}$. Since $a \leq A$ and $b \leq B$, it follows that:

$$\min(a, b) \leq \min(A, B) \implies S_{\text{rep}}^{\mathcal{V}}(Z_1, Z_2) \leq S_{\text{rep}}^{\mathcal{W}}(Z_1, Z_2). \tag{41}$$

$\square$

## A.9. Proof of Proposition 4.7

**Proposition 4.7** (Granular Similarity $\Rightarrow$ Coarser Similarity under $\mathcal{V}$). Let $Z_1$ and $Z_2$ be functionally similar under predictive family $\mathcal{V}$ with respect to task $Y$. Let $Y' = g(Y)$ be a coarser task. If the family $\mathcal{V}$ is closed under post-composition with $g$ (i.e., $\forall h \in \mathcal{V}, g \circ h \in \mathcal{V}$), then $Z_1$ and $Z_2$ are functionally similar under $\mathcal{V}$ with respect to $Y'$.

*Proof.* We aim to show that functional similarity on $Y$ implies functional similarity on $Y' = g(Y)$. By Definition 4.1, functional similarity on $Y$ requires that the usable information gap is zero in both directions:

$$I_{\mathcal{V}}(Z_2 \to Y \mid Z_1) = 0 \quad \text{and} \quad I_{\mathcal{V}}(Z_1 \to Y \mid Z_2) = 0. \tag{42}$$

We will explicitly derive the condition for $Z_2 \to Z_1$; the reverse holds by symmetry.

Using the expansion of usable information from Proposition 4.2, $I_{\mathcal{V}}(Z_2 \to Y \mid Z_1) = 0$ implies that the optimal stitched risk on $Z_1$ is bounded by the native risk on $Z_2$. Assuming $h_2$ is Bayes-optimal for $Z_2$, the stitched predictor cannot strictly outperform it, meaning the expected risks must be exactly equal:

$$\inf_{s \in \mathcal{S}} \mathbb{E}_{z_1}\Big[CE\big(y, (h_2 \circ s)[z_1](y)\big)\Big] = \mathbb{E}_{z_2}\Big[CE\big(y, h_2[z_2](y)\big)\Big]. \tag{43}$$

Let $s^* \in \mathcal{S}$ be the stitcher that achieves this exact infimum.

Now consider the coarser task $Y' = g(Y)$. We define the predictor for $Y'$ as the composition of the original head and the coarsening function $g$. By the closure hypothesis ($\forall h \in \mathcal{V}, g \circ h \in \mathcal{V}$), the composed functions remain valid predictors:

- The head for $Z_2$ becomes $h_2' = g \circ h_2$.

- The stitched predictor for $Z_1$ becomes $h'_{stitch} = g \circ (h_2 \circ s^*)$.

The Cross-Entropy ($CE$) loss on the coarser task $Y'$ is a deterministic function of the prediction on $Y$. Since Equation (43) establishes equality of the expected risks on $Y$, applying the same deterministic function $g$ to both models preserves this equality. Therefore, for the specific stitcher $s^*$:

$$\mathbb{E}_{z_1}\Big[CE\big(y', h'_{stitch}[z_1](y')\big)\Big] = \mathbb{E}_{z_2}\Big[CE\big(y', h'_2[z_2](y')\big)\Big]. \tag{44}$$

Because $h'_{stitch}$ is just one valid predictor within the available stitched family for $Z_1$, the mathematical infimum over *all* possible stitchers $s \in \mathcal{S}$ for task $Y'$ must be less than or equal to this expected risk. Substituting this into Proposition 4.2 for the task $Y'$ gives:

$$I_{\mathcal{V}}(Z_2 \to Y' \mid Z_1) = \max(0, \leq 0) = 0. \tag{45}$$

Because our initial assumption also included $I_{\mathcal{V}}(Z_1 \to Y \mid Z_2) = 0$, applying the exact same derivation in the reverse direction yields $I_{\mathcal{V}}(Z_1 \to Y' \mid Z_2) = 0$. Since both directions equal zero, the representations are functionally similar on $Y'$, strictly satisfying Definition 4.1. □

## A.10. Proof of Corollary 4.8

**Corollary 4.8** (Representational ⇒ Functional Similarity under $\mathcal{V}$). Let $\mathcal{V}$ be a predictive family that is **closed under composition** (i.e., $f, g \in \mathcal{V} \implies f \circ g \in \mathcal{V}$). If $Z_1$ and $Z_2$ are representationally similar under $\mathcal{V}$ (losslessly mappable via $\mathcal{V}$), then they are functionally similar under $\mathcal{V}$ with respect to any deterministic task $Y$ derived from $X$.

*Proof.* We aim to show that functional similarity under $\mathcal{V}$ holds, which by Definition 4.1 requires both $I_{\mathcal{V}}(Z_2 \to Y \mid Z_1) = 0$ and $I_{\mathcal{V}}(Z_1 \to Y \mid Z_2) = 0$. We explicitly derive the condition for $Z_2 \to Z_1$; the reverse holds by symmetry.

According to Proposition 4.2, the usable conditional information is exactly the non-negative gap between the optimal stitched risk and the native original risk:

$$I_{\mathcal{V}}(Z_2 \to Y \mid Z_1) = \max\left(0, \inf_{s \in \mathcal{S}} \mathbb{E}_{z_1}\Big[CE\big(y, (h_2 \circ s)[z_1](y)\big)\Big] - \mathbb{E}_{z_2}\Big[CE\big(y, h_2[z_2](y)\big)\Big]\right). \tag{46}$$

Assume $Z_1$ and $Z_2$ are representationally similar under $\mathcal{V}$. By definition (in the lossless limit), this guarantees the existence of a deterministic, invertible transformation mapping between the two spaces within the accessible family. Let this mapping be $\psi \in \mathcal{V}$ such that $Z_2 = \psi(Z_1)$.

Because $\mathcal{S}$ represents the family of accessible stitchers derived from $\mathcal{V}$, we select this mapping as our candidate stitcher ($\psi \in \mathcal{S}$). Substituting this specific candidate, we note that the composite function evaluated pointwise on $z_1$ is strictly equivalent to the native head evaluated on $z_2$:

$$(h_2 \circ \psi)[z_1](y) = h_2[\psi(z_1)](y) = h_2[z_2](y). \tag{47}$$

Because this pointwise equality guarantees identical expected risks, and $\psi$ is an accessible stitcher in $\mathcal{S}$, the optimal stitched risk (the infimum) cannot exceed the native risk:

$$\inf_{s \in \mathcal{S}} \mathbb{E}_{z_1}\Big[CE\big(y, (h_2 \circ s)[z_1](y)\big)\Big] \leq \mathbb{E}_{z_2}\Big[CE\big(y, h_2[z_2](y)\big)\Big]. \tag{48}$$

Substituting this bound directly into Equation 46 forces the difference to be $\leq 0$, immediately yielding $I_{\mathcal{V}}(Z_2 \to Y \mid Z_1) = 0$.

Symmetrically, representational similarity guarantees the existence of the inverse mapping $\psi^{-1} \in \mathcal{V}$. Applying the exact same derivation for the reverse direction yields $I_{\mathcal{V}}(Z_1 \to Y \mid Z_2) = 0$. Since the gap is zero in both directions, $Z_1$ and $Z_2$ strictly satisfy the conditions for functional similarity under $\mathcal{V}$. □

# B. Architectures Details

*Table 1.* Linear model architectures. All hidden layers use Linear + BatchNorm + ReLU + Dropout(0.1).

| Model | Hidden sizes | # Linear layers | Notes |
|---|---|---|---|
| linear_small | [512, 256] | 3 | Output layer is Linear to num_classes |
| linear_medium | [1024, 512, 256] | 4 | ,,,, |
| linear_large | [2048, 1024, 512, 256] | 5 | ,,,, |
| linear_deep | [512, 512, 256, 256, 128] | 6 | ,,,, |
| linear_wide | [2048, 1024] | 3 | ,,,, |

*Table 2.* Custom CNNs for 32x32 input. All conv blocks use Conv3x3 + BatchNorm + ReLU, with MaxPool2x2 where shown.

| Model | Channels per block | Downsampling | Classifier |
|---|---|---|---|
| tiny_cnn | [16, 32, 64, 128] | MaxPool after first 3 blocks (32→16→8→4) | Linear(128*4*4 → num_classes) |
| narrow_cnn | [32, 64, 128, 256] | MaxPool after first 3 blocks (32→16→8→4) | Linear(256*4*4 → num_classes) |
| narrow_cnn_wide | [64, 128, 256, 512] | MaxPool after first 3 blocks (32→16→8→4) | Linear(512*4*4 → num_classes) |
| simple_cnn | [64, 128] | MaxPool after each block (32→16→8) | Linear(128*8*8 → num_classes) |

*Table 3.* ResNet-20 for CIFAR (basic blocks, no bottleneck). Depth = 6n+2 with n=3.

| Stage | Output size | Details |
|---|---|---|
| Input | 32x32 | 3 channels |
| Conv1 | 32x32 | 3x3, 16 filters, stride 1, BN, ReLU |
| Stage 1 | 32x32 | 3 BasicBlocks, 16 channels, stride 1 |
| Stage 2 | 16x16 | 3 BasicBlocks, 32 channels, first block stride 2 |
| Stage 3 | 8x8 | 3 BasicBlocks, 64 channels, first block stride 2 |
| Pool + FC | 1x1 | Global avg pool, FC(64 → num_classes) |

*Table 4.* DenseNet-40 for CIFAR (L=12 per block, k=12, compression=0.5).

| Stage | Output size | Details |
|---|---|---|
| Input | 32x32 | 3 channels |
| Conv1 | 32x32 | 3x3, 2k=24 filters, stride 1 |
| Dense Block 1 | 32x32 | 12 layers, each BN-ReLU-Conv3x3(k=12) |
| Transition 1 | 16x16 | BN-ReLU-Conv1x1 + AvgPool2x2, compression 0.5 |
| Dense Block 2 | 16x16 | 12 layers, BN-ReLU-Conv3x3(k=12) |
| Transition 2 | 8x8 | BN-ReLU-Conv1x1 + AvgPool2x2, compression 0.5 |
| Dense Block 3 | 8x8 | 12 layers, BN-ReLU-Conv3x3(k=12) |
| Head | 1x1 | BN-ReLU + Global AvgPool + FC(num_channels → num_classes) |

*Table 5.* ShuffleNetV2 adapted for CIFAR (32x32). Stage repeats = [4, 8, 4].

| Stage | Blocks | Output size | Width 0.5 | Width 1.0 | Width 1.5 | Width 2.0 |
|---|---|---|---|---|---|---|
| Stem | Conv3x3, stride 1 | 32x32 | c1=24 | c1=24 | c1=24 | c1=24 |
| Stage2 | 1 downsample (s=2) + 3 normal blocks | 16x16 | c2=48 | c2=116 | c2=176 | c2=244 |
| Stage3 | 1 downsample (s=2) + 7 normal blocks | 8x8 | c3=96 | c3=232 | c3=352 | c3=488 |
| Stage4 | 1 downsample (s=2) + 3 normal blocks | 4x4 | c4=192 | c4=464 | c4=704 | c4=976 |
| Head | Conv1x1 to 1024; GAP; FC 1024→C | 1x1 | 1024 | 1024 | 1024 | 1024 |

*Table 6.* MobileNetV3 adapted for CIFAR (32x32). Width multiplier scales all block output channels and the last conv.

| Stage | Blocks (k, exp, c, SE, s) | Output size | Notes |
|---|---|---|---|
| Stem | Conv3x3, 16 ch, s=1 + BN + HardSwish | 32x32 | CIFAR stem (stride 1) |
| B1 | (3,1,16, yes,1) | 32x32 | Inverted residual |
| B2 | (3,4,24, no,2) | 16x16 | Downsample |
| B3 | (3,3,24, no,1) | 16x16 | |
| B4 | (5,3,40, yes,2) | 8x8 | Downsample |
| B5 | (5,3,40, yes,1) | 8x8 | |
| B6 | (5,3,40, yes,1) | 8x8 | |
| B7 | (5,6,48, yes,1) | 8x8 | |
| B8 | (5,6,48, yes,1) | 8x8 | |
| B9 | (5,6,96, yes,2) | 4x4 | Downsample |
| B10 | (5,6,96, yes,1) | 4x4 | |
| B11 | (5,6,96, yes,1) | 4x4 | |
| Head | 1x1 Conv to 576 ch; GAP; FC 576$\to$1024; Dropout(0.2); FC 1024$\to$C | 1x1 | Classifier |

## C. Scale of Empirical Validation

To ensure the robustness and generalizability of our theoretical claims regarding functional similarity, we conducted an exhaustive empirical evaluation across diverse architectures, capacities, and datasets. Rather than limiting our analysis to a single architecture family or a handful of cherry-picked layers, we evaluated cross-architecture stitching across a fully connected $N \times M$ grid of models.

### C.1. Macro-Architecture Layer Extraction

A naive pairwise evaluation of every individual computational operation (e.g., every `Conv2d` or `BatchNorm` layer) between two deep networks is both computationally intractable and theoretically uninformative, as intermediate operations within a residual block do not represent stable macro-features.

Consequently, we designed a dynamic extraction framework that isolates strictly the macro-architectural checkpoints. For convolutional networks, we restricted our analysis to the outputs of major topological blocks (e.g., the termination of stages in ResNets, or dense block outputs in DenseNets). For purely linear networks, we extracted the representations following each discrete fully connected layer.

The exact number of extracted macro-layers for each evaluated architecture, following the nomenclature established in Appendix B, is as follows:

- **Linear Networks:** `linear_small` (3), `linear_wide` (3), `linear_medium` (4), `linear_large` (5), `linear_deep` (6).

- **Small-Scale ConvNets (CIFAR/MNIST/SVHN/TinyImageNet):** `simple_cnn` (2), `narrow_cnn` (4), ShuffleNetV2 (5), DenseNet-40 (7), ResNet-20 (10), MobileNetV3 (13).

- **ImageNet-Scale Models:** ResNet-18 (9), ResNet-34 (17), ResNet-50 (17).

### C.2. Comprehensive Experimental Grid

We evaluated every valid permutation of macro-layers for a given model pair. For example, comparing a ResNet-20 (10 layers) to a MobileNetV3 (13 layers) requires the optimization and evaluation of 130 distinct stitching transformations for that single model pairing.

By executing a fully connected grid of model comparisons across multiple vision datasets, we effectively mapped the continuous representational landscape of these architectures. In total, we executed 314 distinct model-to-model pairings, resulting in the optimization and evaluation of exactly **12,459 distinct layer-to-layer stitchers**.

The complete scale of this experimental matrix is detailed in Table 7.

*Table 7.* Comprehensive breakdown of the architectures, macro-layers, datasets, and cross-model stitching combinations evaluated in our empirical study. The total number of model-to-model pairs evaluated is 314, yielding exactly 12,459 unique stitching optimizations.

| Experimental Grid | Architectures (Macro-Layers) | Datasets | Model Pairs | Layer Pairs |
|---|---|---|---|---|
| **Linear Networks** | `linear_small` (3), `linear_wide` (3), `linear_medium` (4), `linear_large` (5), `linear_deep` (6) | CIFAR-10, CIFAR-100, MNIST, SVHN, TinyImageNet | 125 | 2,205 |
| **Small-Scale ConvNets** | `simple_cnn` (2), `narrow_cnn` (4), ShuffleNetV2 (5), DenseNet-40 (7), ResNet-20 (10), MobileNetV3 (13) | CIFAR-10, CIFAR-100, MNIST, SVHN, TinyImageNet | 180 | 8,405 |
| **ImageNet Scale** | ResNet-18 (9), ResNet-34 (17), ResNet-50 (17) | ImageNet | 9 | 1,849 |
| | | **Totals:** | **314** | **12,459** |

## D. Regularizers for Stitchers and Similarity Metrics

**Stitcher training (implementation details).** For MLP encoders we use affine linear stitchers $y = Wx + b$; for CNNs we use $1 \times 1$ convolutional stitchers $y = W * x + b$ (optionally followed by adaptive pooling if spatial sizes differ). Stitchers are trained by minimizing the same objective as the task head: cross-entropy by default, or KL-divergence to the target model when using distillation mode. For the orthogonal and orthogonal+scale families we add an explicit orthogonality penalty

$$\|Q^T Q - I\|_F^2$$

(with weight 0.1) on the stitcher's weight matrix $Q$. The orthogonal family uses $y = Qx$ (no scaling), while orthogonal+scale uses $y = sQx$ with a learned scalar $s > 0$. Affine stitchers are unconstrained (no regularization). Invertible-affine stitchers are parameterized as $W = \exp(A)$, which enforces invertibility by construction (no additional penalty in training).

*Table 8.* Regularizers used in stitcher training.

| Method | Regularizer | Weight / Params |
|---|---|---|
| Affine (FC/Conv) | none | – |
| Orthogonal (FC/Conv) | $\|Q^T Q - I\|_F^2$ | 0.1 (no scaling; $y = Qx$) |
| Orthogonal+Scale (FC/Conv) | $\|Q^T Q - I\|_F^2$ | 0.1 + scalar $s$ (uniform scale; $y = sQx$) |

**Similarity metrics (implementation details).** We report standard representational metrics (CKA, RSA, SVCCA, CCA) computed on layer activations; these require no optimization or regularization. For alignment-based metrics, we fit a linear map from source to target features and calculate Equation 15.

For *affine* similarity we minimize MSE directly (linear regression for 2D features, or a learned $1 \times 1$ conv for 4D features). For the *orthogonal* and *orthogonal+scale* similarity families, the alignment uses an orthogonal map $Q$, with the latter also fitting a scalar $s$; these are closed-form in the linear case and optimized with the same orthogonality penalty ($\|Q^T Q - I\|_F^2$, weight 0.1) for the conv-based alignment. For *invertible-affine* similarity we fit $Y \approx XW + b$ and add a singular-value floor penalty (weight 0.01, floor $10^{-3}$) to discourage near-singular transforms. Spatial activations are aggregated before similarity computation using one of {global average pooling, flatten, spatial-samples}.

*Table 9.* Regularizers used in similarity metric fitting.

| Method | Regularizer | Weight / Params |
|---|---|---|
| Affine Linear | none | – |
| Orthogonal Linear (Procrustes) | none (closed-form) | $Y \approx XQ$ (no scaling) |
| Orthogonal+Scale | none (closed-form) | $Y \approx sXQ$ (uniform scale) |
| Affine (Conv 1x1) | none | – |
| Orthogonal (Conv 1x1) | $\|Q^T Q - I\|_F^2$ | 0.1 (no scaling) |
| Orthogonal+Scale (Conv 1x1) | $\|Q^T Q - I\|_F^2$ | 0.1 (with scalar $s$) |

