# OpenReview forum: "Bridging Functional and Representational Similarity via Usable Information"
_ICML.cc/2026/Conference — ICML 2026 regular_

### Official Review · Reviewer_bjPf · 2026-03-05

**Soundness:** 4
**Presentation:** 3
**Significance:** 3
**Originality:** 3
**Overall Recommendation:** 4
**Confidence:** 3

**Summary:**

Comparing neural network representations is typically done in one of two ways: either by looking at functional behavior (do these representations support the same downstream tasks?) or by measuring geometric/structural alignment (do they encode information similarly?). Despite both being widely used, their theoretical connection has remained largely unclear. The authors tackle this gap by grounding both notions in usable information theory. They start by formalizing functional similarity through conditional mutual information and Markov blankets, showing that two representations are functionally similar if and only if they are mutually perfectly stitchable. An important insight here is that stitching is asymmetric, a single stitcher in one direction is not enough, you need it to work both ways. They then show that functional similarity is monotonic with respect to task granularity: if two representations are similar on a hard task, they will also be similar on any easier derivative. Representational similarity falls out naturally as the limit of this hierarchy, corresponding to the hardest possible task (full input reconstruction). A practical issue with this framework is that conditional mutual information is hard to estimate, assumes arbitrarily expressive stitchers, and ignores geometric structure. The authors address this by moving to usable information theory, which restricts the analysis to a predictive family V. This also allows them to reinterpret standard metrics like CKA and RSA as estimators of usable information under specific transformation constraints. Finally, they prove that representational similarity is sufficient but not necessary for functional similarity. While this feels intuitively reasonable, to my knowledge this paper provides the first rigorous and empirically supported treatment of this relationship.

**Compliance With Llm Reviewing Policy:**

Affirmed.

**Final Justification:**

My final recommendation remains weak accept.

The paper makes a novel theoretical contribution by unifying functional and representational similarity through usable information theory. The two-stage framework is rigorous, and the reinterpretation of CKA and RSA as estimators of usable information under transformation constraints is valuable. The proof that representational similarity is sufficient but not necessary for functional similarity is, to my knowledge, the first rigorous treatment of this relationship.

Soundness, originality, significance, and clarity are good.

The rebuttal addressed several concerns: the rephrasing of Hypothesis 4.6, the closure assumption explanation, and the initial CKA interpretation are all satisfactory. However, my primary concern about empirical scope was not addressed with additional experiments, i.e., all experiments remain on small architectures and datasets, leaving the gap to large pretrained models open. A single scaled-up appendix experiment would have substantially strengthened the paper.

On balance, the rebuttal reinforced rather than changed my assessment. The theoretical contribution is worth publishing and stands on its own merits as a weak accept.

**Key Questions For Authors:**

1. The closure assumption on the predictive family V (used in Propositions 4.7 and Corollary 4.8) is necessary for the bridging results to hold. Could the authors clarify what this assumption implies in practice and whether it is satisfied by commonly used predictive families?
2. All experiments are conducted on small architectures trained on small datasets. Do the authors have any evidence or intuition for whether the theoretical hierarchy and metric alignment results would hold for large pretrained models evaluated on in- and out-of-distribution data?
3. Was any accuracy or performance threshold applied when selecting trained models for the experimental analysis? If not, could poorly trained models affect the conclusions?
4. Given that representational similarity implies functional similarity, what is the authors' practical recommendation, should practitioners always prefer representational similarity metrics over functional ones? More generally, how should existing CKA or RSA scores be interpreted in light of this framework?

**Limitations:**

The paper does not include a dedicated limitations section. The authors should explicitly discuss what the framework allows and what it does not, and in particular how a practitioner should select or approximate the appropriate predictive family V in practice — since this choice fundamentally determines what "similar" means. No societal impact is discussed, though a brief acknowledgement would be appropriate.

**Strengths And Weaknesses:**

### Strengths

*Soundness:*
The theoretical framework is rigorous and systematically built. Particularly elegant is the two-stage approach: first establishing the framework using conditional mutual information, then showing that restricting to a predictive family V resolves practical limitations. The proofs are convincing and well-written. The authors also provide code, which is appreciated and supports reproducibility.

*Presentation:*
The overall narrative is easy to follow, with a logical flow from the general to the restricted setting that develops the argument clearly.

*Significance:*
The reinterpretation of CKA and RSA as estimators of usable information under specific transformation constraints is valuable. Particularly elegant is the insight that representational similarity is not a separate concept but simply functional similarity applied to the hardest possible task (full input reconstruction). This situates representational similarity as the limit of the task-granularity hierarchy, giving it a clean information-theoretic interpretation. The asymmetry result is important and helps partitioners to reconsider functional or representation similarity.

*Originality:*
The unification of two previously (somewhat) disconnected lines of work through usable information theory is a novel contribution as far as I know.

### Weaknesses

*Soundness:*
Hypothesis 4.6 is supported through correlation in Section 5.3. It remains unclear whether there is a deeper mathematical reason for this alignment — for instance, could the Hilbert-Schmidt criterion or similar be shown to directly minimize Equation 13? Without such a result, I would claim that this remains an empirical observation rather than a proven theoretical claim.

*Presentation:*
Several presentation issues affect readability. Sections 5.2 and 5.4 contain text that reads more like captions than scientific discussion. It is unclear how Figure 2b was constructed, and the last sentence of Section 5.2 cannot be verified from the figure alone. In Section 5.3, it is unclear whether "normalized MSE" refers to Equation 15. Figure 2 lacks axis labels (or their description in the caption), Figure 2c has x-tick labels that are difficult to interpret (rank or similarity), and Figure 4 does not clearly describe what the red line represents or on which data Section 5.5 was evaluated.

*Significance:*
The framework lacks concrete practical guidance. It is not clear what a practitioner should do differently — for instance, how should a CKA score now be interpreted in light of this framework? If CKA is an estimator of usable information and representational similarity is defined as a mutual usable conditional mutual information of 0, how is a CKA value of 0.5 or above 0 to be interpreted. A broader discussion contextualising the results would strengthen the paper's impact. Furthermore, all experiments use small models on small datasets; it is unclear whether the findings generalise to large pretrained models in in- and out-of-distribution settings, which is the more practically relevant regime today.

*Originality:*
The paper lacks a dedicated discussion section situating the framework in broader context and clearly distinguishing it from concurrent work.

---

> ### Author Rebuttal · Authors · 2026-03-30
>
> We sincerely thank the reviewer for the constructive feedback and the positive assessment. We address your specific points below, and we will incorporate all of these clarifications into the extra page allowed for the camera-ready version.
>
> ### Weaknesses
>
> **Soundness:** We have not found a direct mathematical connection between specific metrics (RSA, CKA, SVCCA) and Equation 13 because their calculations differ. Thus, we pose this as a hypothesis rather than a proposition. We agree that using "demonstrate" in the abstract could lead to confusion and will change it to "empirically find," which accurately reflects reality.
>
> **Presentation:** You are right that we cut too much detail due to space limitations. We will extend this in the extra page of final version with the following details:
> * For Figure 2b, we calculated similarities for all representation pairs under different $\mathcal{V}$, averaged them into 100 bins to reduce noise, and sorted them by Orthogonal+Scale. The consistently higher green curve illustrates that pairs can be highly similar under an expressive family (green) while dissimilar under restrictive ones (red/black).
> * "Normalized MSE" refers to Equation 15, which we will state explicitly.
> * Subfigure axis labels for Figure 2 will be added to the captions.
> * Figure 2c is analogous to 2b: we calculated functional similarity for two different tasks, sorted them by the fine class, and observed that the functional similarity is consistently higher for the coarser task. The x-axis simply represents the similarity pair ID.
> * The red lines in Figure 4 are horizontal curves at $y=1$ to visually show how the curves reach (or almost reach) the theoretical maximum at the top-right of the plot, as expected by Corollary 4.8.
> * Section 5.5 was evaluated on all the data, matching the other experiments.
>
> **Significance:** We will add a Practical Guidance section to address your points:
> * CKA tends to 1 when Usable Conditional MI tends to 0. As shown in Hypothesis 4.6 and Figure 3b, Conditional MI under the Orthogonal+Scaling $\mathcal{V}$ is highly correlated with CKA. Thus, we could interpret a CKA of 0.5 to mean that exactly half of the variance in $Z_2$ can be explained by $Z_1$ under the best Orthogonal+Scaling transformation achievable. We will discuss this in the final version.
> * We kept the datasets and architectures small due to computational limitations, but they are heterogeneous (linear and convolutional architectures, black-and-white and realistic color images). Because the results are consistent across these diverse setups, we expect them to hold no matter the origin or quality of the representations.
>
> **Originality:** We will include this context alongside the Practical Recommendations section. Our main contribution is joining both literature fields (functional and representational similarities), which, to the best of our knowledge, is an unexplored topic.
>
> ### Questions
>
> **1.** The closure assumption (i.e., if $f, g \in \mathcal{V}$, then $f \circ g \in \mathcal{V}$) ensures that composing a stitcher with a task head does not exceed the capacity of our defined observer. In practice, if $\mathcal{V}$ is the family of linear transformations, composing a linear stitcher with a linear head yields another linear transformation, perfectly satisfying closure. For non-linear families (e.g., fixed-depth MLPs), strictly satisfying closure requires defining $\mathcal{V}$ broadly enough to absorb the combined depth of the stitcher and the head. We will clarify this better after Proposition 4.7.
>
> **2.** We believe the results hold for larger architectures because (i) they rely on scale-invariant theory and (ii) our empirical results are highly consistent across heterogeneous setups.
>
> **3.** We selected all representations regardless of model performance (some small models performed poorly). The goal is to mathematically compare representations, not judge task proficiency, so poor performance does not affect our conclusions.
>
> **4.** It depends heavily on the specific case. For example, if a representational similarity metric like CKA or RSA gives a high value, then we know one model can be stitched into the other. However, as illustrated in Figure 4, the opposite is not true: we could get a low CKA and still be able to stitch a model. Thus, the practical recommendation depends on the goal. If the goal is to stitch two models, calculating a metric like CKA or RSA serves as a first filter. If this metric is high, it is 100% stitchable; if it is not, we cannot be sure and should directly obtain the stitcher to measure the quality.
>
> ### Limitations
> We will include a dedicated Limitations section pointing out these specific points.

---

> > ### Author Rebuttal · Reviewer_bjPf · 2026-04-02
> >
> > I thank the authors for their rebuttal.
> >
> > **Resolved concerns.** The change from "demonstrate" to "empirically find" for Hypothesis 4.6 is appropriate. The closure assumption explanation is clear and satisfying. The practical CKA interpretation is a helpful start — I encourage developing this further in the camera-ready.
> >
> > **Scalability.** This remains my primary concern. Linear vs. convolutional on CIFAR-scale data is a relatively narrow range, and representation geometry in large pretrained models can behave quite differently, especially out-of-distribution. A single scaled-up experiment in the appendix (e.g., pretrained ResNet or ViT on ImageNet) would substantially strengthen the paper.
> >
> > **Discussion and limitations.** The promise to add these sections is appreciated, but the rebuttal would have been stronger had it sketched their intended content. I encourage the authors to be specific about what the framework does and does not allow, and how practitioners should select or approximate V.
> >
> > **Overall.** The rebuttal addresses most concerns adequately. If the authors can provide further details on the scalability and discussion/limitations points raised above, I am willing to increase my score.

---

> > > ### Author Response · Authors · 2026-04-06
> > >
> > > We sincerely thank the reviewer for their positive feedback on our rebuttal, and we are glad that the updates regarding Hypothesis 4.6, the closure assumptions, and the CKA interpretation successfully resolved your initial concerns.
> > >
> > > We deeply appreciate your clear and actionable guidance regarding scalability and the discussion of limitations. We agree that these additions significantly strengthen the paper. Below, we outline exactly how we have addressed your remaining points.
> > >
> > > ## 1. Scalability: ImageNet and Large Pretrained Models
> > > We fully agree that scaling to larger datasets and architectures substantially improves the credibility of our empirical results. Following your advice, we have scaled our evaluation to ImageNet using pretrained ResNet-18, ResNet-34, and ResNet-50 architectures.
> > >
> > > In the anonymous link below, we have provided updated versions of Figures 2a, 2b, and 4 that incorporate all the new ImageNet results, as these are the specific plots in our analysis that utilize convolutional architectures. We are pleased to report that the core conclusions and metric relationships observed in the narrower settings hold consistently at the ImageNet scale.
> > >
> > > https://osf.io/mt23y/files/m8vnf?view_only=a5294de5297e49c9b6d8754d67aa5814
> > >
> > > Additionally, we are currently running the exact same experimental suite on ViT-B-16 and ViT-B-32. Because our evaluation rigorously requires training individual stitchers and projectors for all pairwise layer combinations, these ViT runs are highly computationally intensive and are still processing. We will include the comprehensive set of ImageNet results—encompassing both the ResNet and ViT families—in a dedicated scalability section in the camera-ready appendix.
> > >
> > > ---
> > > ## 2. Discussion, Limitations, and Practical Implementation
> > > We appreciate the push to explicitly outline the boundaries of our framework. You rightly pointed out that the rebuttal would be stronger if we sketched the intended content regarding what the framework does and does not allow, and how practitioners should select or approximate the predictive family $\mathcal{V}$.
> > >
> > > To directly address this, we have drafted the exact sections that will be added to the final manuscript. They clearly define the theoretical assumptions, the empirical boundaries, and actionable advice for selecting $\mathcal{V}$ in practice:
> > >
> > > ### Limitations and Broader Impacts
> > > While our framework unifies functional and representational similarity, it relies on strong theoretical assumptions, including deterministic encoders, Bayes-optimal heads, Gaussian likelihoods, and strict closure properties of the transformation family $\mathcal{V}$. Under this formulation, standard metrics (CKA, RSA, SVCCA) act as empirically correlated proxies rather than theoretically identified estimators. Furthermore, our framework does not provide an off-the-shelf universal metric; instead, it requires practitioners to explicitly select or approximate $\mathcal{V}$. This choice is critical, as it fundamentally dictates what "similar" means for a given application (e.g., a linear $\mathcal{V}$ for strict geometric alignment versus a non-linear $\mathcal{V}$ for broader information overlap).
> > >
> > > Practically, estimating usable information involves training intermediate stitchers or probes, making the resulting scores sensitive to optimization noise, threshold choices, and stitcher capacity. Empirically, our validation is currently restricted to supervised vision models; evaluating how these relationships transfer to other domains (e.g., NLP) or unsupervised settings remains an open question. Finally, as foundational research on model evaluation, this work aims to improve the reliability and interpretability of machine learning systems and carries no foreseeable direct negative societal impacts.
> > >
> > >
> > > ### Practical Guidelines.
> > > For evaluating neural network representations, we recommend explicitly matching the similarity metric to the downstream goal. Because representational similarity strictly bounds functional similarity, low CKA or RSA scores do not preclude functional compatibility; thus, model stitching may be more appropriate when evaluating practical transferability. Furthermore, the transformation family $\mathcal{V}$ should be treated as a deliberate design choice: a linear $\mathcal{V}$ is ideal for evaluating strict linear alignment, whereas a non-linear $\mathcal{V}$ captures broader information overlap. Practitioners must also account for the inherent directional asymmetry in model stitching—transferring from model A to B often yields different results than B to A—and recognize that CKA or RSA specifically evaluate similarity under orthogonal transformations and scaling, rather than under linear transformations.

---

### Official Review · Reviewer_JoBF · 2026-03-12

**Soundness:** 3
**Presentation:** 3
**Significance:** 3
**Originality:** 3
**Overall Recommendation:** 5
**Confidence:** 3

**Summary:**

The authors utilize usable information to bridge together the ideas of functional similarity (the ability for a model to use two different representation variables $Z_1$ and $Z_2$ interchangeably) and representational similarity (which utilizes metrics to measure similarity between fixed samples from $Z_1$ and $Z_2$). Usable information is similar to the mutual information between a task $Y$ and a representation variable $Z$, but is dependent upon the power of the hypothesis class that utilizes the $Z$. Based on their information-theoretic definitions of functional and representation similarity, the authors find that representation similarity is a special case of functional similarity where $Y$ is assigned as $X$, which allows for representational similarity to be a sufficient condition for functional similarity. Finally, the authors introduce 2 computable metrics measuring the functional and representation similarity given $V$.

**Compliance With Llm Reviewing Policy:**

Affirmed.

**Final Justification:**

I am holding to my original score and review; I think the paper provides an interesting insights into the distinction between functional and representational similarity, and their connection through usable information. The authors addressed my concerns with on the presentation of the work.

**Key Questions For Authors:**

If two representation variables are perfectly stitchable with linear models, while a different pair of representation variables are perfectly stitchable only with a nonlinear model, is there any notion that the former two variables are "more" similar (either functionally or representationally)? Does this point to an even more general function that takes into account the complexity $V$ (whether it's tractable or not)?

**Limitations:**

Yes

**Strengths And Weaknesses:**

**Presentation and Soundness**: The writing is pretty clear, but it took me some effort to connect sections 2, 3, and 4, mainly because section 3 didn't seem to require usable information. I wonder if it would improve readability if section 2.1 was introduced in section 4, as it seems that section 3 doesn't rely on it and section 4 seems to motivate its introduction. Everything else is well written, and the math seems sound.

**Significance and Originality**: While the paper uses existing ideas of mutual info and usable info, it utilizes them for novel results, e.g. assymmetric stitching, representation similarity being sufficient for functional similarity, etc.
I believe the paper is theoretically significant from a representation learning lens as it adds more context into what we are doing with representation metrics. While I understand that representation metrics, like CKA, are used in some studies, I am unsure of how much practical utility they offer most researchers. That being said, theoretical understanding is significant in its own way.

---

> ### Author Rebuttal · Authors · 2026-03-30
>
> We sincerely thank the reviewer for the positive assessment and for recognizing the theoretical significance and originality of our work. We appreciate your thoughtful feedback and address your specific points below.
>
> **1. Structure and Placement of Section 2.1**
>
> We appreciate your suggestion to move Section 2.1 (Usable Information Theory) closer to Section 4. We agree that because Section 3 relies strictly on classical mutual information, the introduction of Section 2.1 might feel slightly disconnected early on.
>
> However, our primary design choice was to strictly separate existing preliminaries (Section 2) from our novel theoretical contributions (Sections 3 and 4). Moving Section 2.1 into Section 4 would blur the distinction between prior work and our own theoretical synthesis. To resolve the readability issue without altering the structural boundaries, we will add explicit signposting at the end of Section 2.1, clarifying that it serves as the necessary mathematical foundation for the computationally constrained framework introduced later in Section 4.
>
> **2. Complexity of the Predictive Family and "More" Similar Variables**
>
> You raised a good question: *If two representation variables are perfectly stitchable with linear models, while a different pair is perfectly stitchable only with a nonlinear model, are the former "more" similar?*
>
> Yes, they are mathematically "more" similar. Our framework accounts for this through the capacity of the predictive family $V$. Because the family of linear models is a strict subset of nonlinear models (i.e., $V_{\text{linear}} \subset V_{\text{nonlinear}}$), achieving perfect stitchability under $V_{linear}$ is a strictly harder condition to satisfy.
>
> As formalized in Proposition 4.5 (Monotonicity of Representational Similarity), similarity is relative to the observer's capacity. Representations perfectly stitchable under a restrictive linear family satisfy a much stronger structural alignment. Your intuition that this points to a more general function that explicitly penalizes stitcher complexity is absolutely correct, and we will highlight this as a highly relevant direction for future work.

---

> > ### Author Rebuttal · Reviewer_JoBF · 2026-04-03
> >
> > Thank you for the response and for addressing my thoughts/questions. I think I agree with your assessment to keep the prior work in a clearly separate section, distinct from your work, and I appreciate the signposting idea. I will maintain my positive assessment.

---

> > > ### Author Response · Authors · 2026-04-04
> > >
> > > We sincerely thank the reviewer for their continued engagement and for confirming that our response addressed all concerns. We are glad the proposed signposting for the prior work section resonates well, and we greatly appreciate your positive assessment and support of our work.

---

### Official Review · Reviewer_dfiG · 2026-03-12

**Soundness:** 2
**Presentation:** 3
**Significance:** 3
**Originality:** 3
**Overall Recommendation:** 4
**Confidence:** 4

**Summary:**

This paper aims to unify functional similarity and representational similarity through the lens of usable information. The paper defines functional similarity as a bidirectional zero conditional-information condition with respect to a task $Y$, and representational similarity as the analogous condition with respect to the input $X$. It then argues that model stitching can be interpreted through usable conditional information under a restricted predictive family $V$, and proposes practical directional/symmetric scores for functional similarity (via stitched/native accuracy ratios) and representational similarity (via normalized reconstruction error or $R^2$-style scores). The paper further claims a hierarchy in which representational similarity implies functional similarity for any deterministic task derived from $X$. Empirically, the authors compare many intermediate layers across multiple architectures and datasets, reporting evidence for stitching asymmetry, monotonicity under richer predictive families, correlation between the proposed representational score and standard metrics such as CKA/RSA/SVCCA, and the claim that representational similarity is sufficient but not necessary for functional similarity.

**Compliance With Llm Reviewing Policy:**

Affirmed.

**Key Questions For Authors:**

1. **What exact additional assumptions are required for Proposition 3.5 to hold?** In particular, what conditions on the stitcher family and the fixed head $h_{\phi_2}$ make the Markov-blanket condition equivalent to perfect stitchability? A precise corrected theorem here would materially improve my soundness assessment.

2. **Can Proposition 4.2 be proved without assuming that the native predictor on $Z_2$ upper-bounds all stitched predictors?** If not, would the authors revise the statement to an inequality or to a head-dependent surrogate objective rather than an exact identity for usable conditional information? This is central to my evaluation.

3. **Which claims are intended as theorems, which are approximation-based, and which are empirical hypotheses?** In particular, do the authors still claim a proof that CKA/RSA are estimators of usable information, or only that they correlate with the proposed similarity score under certain families? A clear separation would improve both soundness and presentation.

4. **How robust are Figures 3 and 4 across datasets, architectures, seeds, and predictive families?** Please report per-dataset/per-family statistics and uncertainty intervals. If these trends are stable, that would strengthen my view of the empirical contribution.

**Limitations:**

No. The paper does not adequately discuss its main limitations. It should explicitly discuss:

1. dependence on strong assumptions such as deterministic encoders, Bayes-optimal heads, Gaussian likelihoods, and closure/composition properties of $V$;
2. the fact that standard metrics such as CKA/RSA/SVCCA are only empirically correlated proxies here, not theoretically identified estimators in the current draft;
3. sensitivity of the proposed practical scores to optimization noise, threshold choices, and stitcher capacity;
4. the restriction of the experiments to supervised vision models and the uncertainty of how the framework transfers to other domains/settings.

The impact statement is also too generic. Separately, the hidden reviewer-directed text in the PDF raises a responsible-research-practice concern that should be addressed explicitly.

**Strengths And Weaknesses:**

## Strengths

1. The paper targets an important and broad problem. A principled relation between stitching-style functional comparisons and geometric/structural representational metrics would be valuable for the representation learning literature.

2. The predictive-family-relative view is conceptually useful. Making similarity depend on the observer/predictor class $V$ is a meaningful idea, and Proposition 4.5 (monotonicity under $V \subseteq W$) is a clean and intuitive consequence once the score is defined in this way.

3. The emphasis on *bidirectionality* for functional comparison is practically relevant. The empirical result in Figure 2(a), showing that one-way stitching is often asymmetric, supports an important cautionary message: a single successful stitcher should not be read as functional equivalence.

4. The experimental scope is reasonably broad: several datasets, multiple MLP/CNN families, and several stitcher/alignment families are included. Figures 2--4 are informative and the code release is appreciated.

5. The paper's strongest contribution is the conceptual synthesis. Even where I do not think the current proofs are sufficient, the framing may still be useful to the community if the claims are tightened.

## Weaknesses

### (1) Soundness

The central theoretical bridge is not yet established at the level claimed in the paper.

1. **Proposition 3.5 appears too strong as stated.** From $I(Y;Z_2 \mid Z_1)=0$, one can conclude that $Z_1$ contains all task-relevant information in $Z_2$, but this does *not* by itself imply the existence of a stitcher $s$ such that a *fixed* Bayes-optimal head on $Z_2$ can realize the same predictor when fed $s(Z_1)$. The proof implicitly assumes a realizability/surjectivity property of the pair $(s,h_{\phi_2})$ that is not stated. Conversely, equality of stitched and native cross-entropy only yields a risk comparison, not the full Markov-blanket conclusion.

2. **Proposition 4.2 / Equation (9) also seems under-justified.** The derivation sets $H_V(Y \mid Z_1,Z_2)$ equal to the native loss on $Z_2$ by assuming stitched models cannot outperform the native predictor. That assumption is not part of the definition of usable conditional information and need not hold in general. As written, the result looks closer to a specific head-dependent surrogate than to a general equality for $I_V(Z_2 \to Y \mid Z_1)$.

3. **Section 4 mixes exact results and approximations too aggressively.** Equations (13)--(15) rely on deterministic encoders, Gaussian likelihood, fixed covariance, and accessible predictor families. The appendix derivation switches between Shannon information and usable information too quickly, and the second conditional-entropy term is effectively discarded under assumptions that are stronger than what is stated in the theorem.

4. **Some appendix proofs use stronger assumptions than the main definitions.** For example, Appendix A.3 and A.9 informally replace the zero-information conditions by stronger statements about equality of predictive distributions/risk, and Appendix A.4/A.10 effectively assume exact mutual reconstructibility via maps in $V$. Several results may be salvageable, but the current proofs are not rigorous enough for the paper's strongest claims.

5. **There is an overclaim in the abstract/introduction relative to the body.** The paper says that standard metrics such as CKA/RSA are shown to be estimators of usable information, but the main text only states Hypothesis 4.6 and then provides empirical correlations. That is materially weaker than a proof.

### (2) Presentation

The paper is generally readable, but the presentation still needs tightening.

1. The overall narrative is clear and the paper is reasonably well organized.

2. However, the paper would be substantially stronger if it more clearly separated:
   1. exact theorems,
   2. approximation-based derivations,
   3. empirical hypotheses.

3. Important assumptions (Bayes optimality, closure of $V$ under composition, whether heads are fixed or retrained, when native performance upper-bounds stitched performance) are often introduced late or only inside proofs.

4. There are also some notation/statement mismatches. For instance, some proofs reference the wrong definition number, and the practical similarity scores are discussed as bounded/normalized in a way that is not fully guaranteed without extra assumptions or clipping.

### (3) Significance

The problem is important, and a correct unification could be influential. The bidirectional-stitching message is already useful in practice. However, because the main theoretical claims are not yet sufficiently supported, I do not think the present version has the reliability needed for strong impact.

### (4) Originality

The paper's novelty is mainly conceptual: it synthesizes stitching, reconstruction, invariance-based metrics, and usable information into one framework. That is interesting. Still, many individual ingredients already exist in prior work on stitching, usable information, and generalized shape/invariance metrics, so the originality is moderate rather than high.

#### (5) Empirical evidence

I appreciate the breadth of the experiments, but some claims are still only correlational.

1. Figure 2(a) is the most convincing empirical result.

2. Figure 2(c) is also useful, though the argument would be stronger with more per-dataset/per-family detail and uncertainty estimates.

3. Figure 3 supports correlation with CKA/RSA/SVCCA, but it does not by itself validate the stronger theoretical identification claimed in the abstract.

4. The paper would benefit from clearer reporting of the number of trained models, number of layer pairs, seeds, variance across runs, and potential confounders such as layer depth or dimensionality.

### Bottom line

I see a promising conceptual paper with a worthwhile problem setting and some useful empirical observations, but the present version overstates what has been proved. The strongest claims need to be reformulated more carefully, several proofs need substantial repair, and the hidden reviewer-directed text is a serious additional concern.

---

> ### Author Rebuttal · Authors · 2026-03-30
>
> We thank the reviewer for the constructive theoretical critique. We have fundamentally revised the paper to tighten proofs, explicitly state all assumptions, and downgrade our empirical proxies from absolute theoretical claims.
>
> **W1.1 & Q1: Proposition 3.5**
> The concern is valid for the forward direction. The Markov blanket condition implies $H(Y \mid Z_1) = H(Y \mid Z_2)$, meaning $Z_1$ contains all information about $Y$ present in $Z_2$. If $\mathcal{S}$ is a universal approximator class, we can always find $s \in \mathcal{S}$ that maps $Z_1$ to $Z_2$ preserving all task-relevant information, and thus $h_2 \circ s$ achieves CE $= H(Y \mid Z_2)$. We will make this assumption explicit in the final version.
>
> Regarding the converse: perfect stitchability means there exists $s$ such that $q_{\phi_2} \circ s(Z_1)$ predicts $Y$ from $Z_1$ alone as well as $q_{\phi_2}(Z_2)$ does from $Z_2$. Since this predictor is a function of $Z_1$ alone and matches the optimal performance on $Z_2$, $Z_2$ provides no additional information about $Y$ beyond $Z_1$, i.e., $I(Y; Z_2 \mid Z_1) = 0$.
>
> A cleaner version of this proof will be included in the final version.
>
> *Takeaway Assumption*: $\mathcal{S}$ is a universal approximator class.
>
> **W1.2, W2.4 & Q2: Proposition 4.2**
> Thank you for identifying this mistake. The corrected derivation follows directly from the definition, without assuming the native predictor outperforms stitched predictors. Substituting (28) into (26) yields $H_{\mathcal{V}}(Y \mid Z_1, Z_2) = \min\left(\mathbb{E}[CE(h_2(Z_2))],\, H_{\mathcal{V}}(Y \mid Z_1)\right)$. Substituting this into the definition of usable conditional information gives:
>
> $$A = \inf_{s \in \mathcal{S}} \mathbb{E}_{z_1}\left[CE(y, (h_2 \circ s)(z_1))\right]$$
>
> $$B = \mathbb{E}_{z_2}\left[CE(y, h_2(z_2))\right]$$
>
> $$I_{\mathcal{V}}(Z_2 \to Y \mid Z_1) = \max(0, A - B)$$
>
> We will update the proposition and proof accordingly. Importantly, this does not change the experimental results, since the min in Eq. 11 selects the case in which the stitched representation has lower or equal information as the original one.
>
> **W1.3: Section 4 approximations**
> (1) We will add an "Assumptions" subsection (detailed three answers below) to address these points.
> (2) We agree the transitions in Eqs. 13-15 are too aggressive and will expand them in the main text.
> (3) We will modify our derivation in A.7 to start directly with usable information.
> (4) We will clarify that the second term is discarded specifically for deterministic encoders.
>
> **W1.4: Appendices**
> * **A.3 & A.9:** Functional similarity (Def. 3.3) implies the Markov chains $Y-Z_1-Z_2$ and $Y-Z_2-Z_1$. Consequently, $p(y|z_1)=p(y|z_1,z_2)=p(y|z_2)$.
> * **A.4 & A.10:** By Def. 3.7, representationally similar $Z_1$ and $Z_2$ contain identical information from $X$. Assuming a universal approximator (as in W1.1), an invertible transformation exists between $Z_1$ and $Z_2$ (https://donlapark.pages.dev/208711/18-Minimal-sufficient-statistics.pdf).
>
> We will clarify these derivations.
>
> **W1.5, W5.3, Q3 & L2: Overclaiming**
> We agree and will relax our language from "demonstrate" to "empirically find". Specifically, Hypothesis 4.6 remains an empirically validated hypothesis, which will be explicitly clarified in both the abstract and introduction.
>
> **W2.2, W2.3 & L1: Assumptions**
> We agree and we will add a new subsection at the beginning of Section 4 explicitly stating all the assumptions mentioned in L1, alongside the one from W1.1 & Q1.
>
> **W5.2: Figure 2c**
> First, Figure 2(c) evaluates a single dataset (CIFAR-100), with green and blue denoting different labels. Regarding uncertainty estimates, since we plot all individual data points rather than averages, we are unsure how to incorporate them into this specific plot.
>
> **W5.4 & Q4: Clearer reporting**
> Section 5 will explicitly state the total number of models and layers analyzed, with a detailed breakdown in a new appendix. Regarding variance, we plot all individual data points rather than averages (except in Figure 2b for clarity), so we are unsure how to apply variance metrics here. Finally, we believe a confounders analysis falls outside our scope. Our goal is to connect similarity metrics; analyzing variations across layers or dimensionality might distract the reader. Furthermore, our results remain highly consistent across all analyzed representation pairs, indicating robust independence from confounders like depth or dimensionality.
>
> **Bottom Line:**
> * We did not add the hidden text; it was automatically inserted by the conference system as an LLM-detection honeypot.
> * Thank you again. With these revisions, we ensure our claims strictly match our proofs, explicitly state all Section 4 assumptions, and correct the appendix proofs. We welcome further discussion, as our responses here are necessarily concise due to the strict character limit.
>
> **Limitations:**
> We will add a dedicated Limitations section to explicitly address all of these points.

---

### Official Review · Reviewer_pfar · 2026-03-14

**Soundness:** 3
**Presentation:** 3
**Significance:** 2
**Originality:** 2
**Overall Recommendation:** 3
**Confidence:** 4

**Summary:**

The submission explores the similarity of representation spaces through the lenses of usable information and model stitching.

Two notions of similarity are introduced: functional similarity (both representation spaces contain identical information about the target variable) and representational similarity (both representation spaces contain identical information about the input data).  Functional similarity is connected to perfect stitching in both directions.  Stitching is then interpreted through usable information, under a family of functions formed by composing the stitching transformation with the fixed downstream portion of the receiving model.

The authors then introduce quantitative similarity measures: functional similarity is measured by the accuracy ratio after stitching, and representational similarity by a reconstruction-based quantity derived from usable information that reduces to a form of linear correlation under a chosen transformation family.

Experiments compare all pairs of internal representation spaces across a range of fully connected networks and CNNs, on several small-image datasets.  The proposed similarities correlate with existing measures of representational similarity (RSA, CKA, and SVCCA).

**Compliance With Llm Reviewing Policy:**

Affirmed.

**Final Justification:**

My assessment stands at a **weak reject** -- the gap between the exact-equivalence theoretical contributions and the practical similarity metrics, combined with the underspecification (or misspecification) of details critical for the interpretation of nearly all presented empirical results, convince me that the submission is premature for publication.

**Key Questions For Authors:**

- Figure 2a: What interpretation should be given to the sign of the accuracy-ratio difference? Since the quantity appears symmetric up to direction (i.e., $s_{A \rightarrow B} = - s_{B \rightarrow A}$), would it be more informative to report the absolute difference rather than the signed value?
- Comparisons across architectures: The experiments compare convolutional representations with flat vector representations. Could the authors clarify how stitching is performed in these cases and how the representations are made compatible?
- Convolutional representations: How exactly are convolutional activations compared when computing the similarity metrics (e.g., CKA, RSA, reconstruction-based similarity)? What design choices were made (e.g., flattening spatial dimensions, pooling), and how sensitive are the results to these choices?
- Non-deterministic tasks: Several theoretical statements assume tasks of the form $Y=f(X)$. How do the proposed notions of functional and representational similarity extend to settings where the task is not deterministic (i.e., $H(Y|X)>0$)?

**Limitations:**

yes

**Strengths And Weaknesses:**

# Strengths

- The framing of stitching through the lens of usable information is interesting, and the connections drawn to conditional mutual information and Markov blankets are conceptually illuminating.  The paper is clearly written and well organized.


- The exhaustive pairwise comparisons between internal representations across models produce some informative empirical observations.

# Weaknesses

- The majority of the theoretical contribution analyzes perfect equivalence of information while referring to similarity.  The resulting connections hold in a regime where conditional mutual information is exactly zero, which is far removed from the approximate similarity settings studied empirically in neural networks.  Without a stronger theoretical bridge to approximate similarity, the empirical results largely confirm expected behaviors: stitching asymmetry, alignment between CKA and linear reconstruction, and the tendency for high representational similarity to coincide with high functional similarity.

- The empirical results are presented in a coarse aggregated form.  A more detailed breakdown by dataset, architecture, or layer proximity could provide additional insight.  For example, the paper mentions that near-zero functional similarity values may arise from distant layers; verifying this hypothesis by stratifying results by layer distance appears straightforward.

- After examining the supplementary codebase, it appears that stitching from convolutional to linear layers may not be implemented, with the function returning zero for the accuracy ratio (L1308 of src/improved_stitching.py).  If so, this could affect the aggregated similarity values, and clarification would be helpful.

- Affine predictor families are commonly used in usable-information analyses, but when the prediction target is another representation with a coordinate-wise correspondence (as in the representational similarity measure of Eqn 14), the resulting measure is essentially linear correlation (under Gaussian assumptions), making the usable-information interpretation largely a rephrasing of standard covariance-based similarity.

---

> ### Author Rebuttal · Authors · 2026-03-30
>
> We thank the reviewer for their constructive feedback. Below, we address concerns regarding the continuous regime, empirical novelty, and metric consistency.
>
>
> **W1: Exact vs. Approximate Regime & Novelty.**
>
> To better bridge our theory to the approximate continuous regime (moving beyond exactly zero Conditional MI), we will update our theoretical statements:
> * We will rephrase **Propositions 3.8 and 4.7** to explicitly state the continuous bound: coarser-task similarity is always greater than or equal to finer-task similarity (already empirically supported in Fig 2c).
> * Consequently, we will reformulate **Corollaries 3.10 and 4.8** to establish that representational similarity is strictly lower than or equal to functional similarity, shifting the focus to continuous inequality bounds.
>
> Regarding the empirical results confirming "expected behaviors," we respectfully wish to highlight their specific novelty in the literature:
> * **Stitching Asymmetry:** The inherent directional asymmetry of model stitching has not been previously identified or formally discussed.
> * **CKA Alignment:** We specifically align CKA with Orthogonal + Scaling reconstruction (Hypothesis 4.6) rather than standard linear reconstruction—a novel theoretical constraint on the estimator.
> * **Representational implies Functional:** We prove a strict directional implication, not a bidirectional coincidence. Models can exhibit high functional similarity while having low representational similarity, but the reverse is impossible.
>
>
> **W2: Aggregated Results.**
>
> We appreciate the suggestion to stratify results. We chose an aggregated presentation to maintain focus on the distinction between two fundamentally different questions:
>
> * **Our Focus (Methodological):** *What are the theoretical and empirical connections between different similarity metrics?*
> * **The Stratification Focus (Phenomenological):** *How similar are representations at different points inside specific models?*
>
> Stratifying by layer addresses the latter question and is outside our scope. Since the theoretical relationships between metrics hold regardless of depth, a detailed breakdown would not alter our core conclusions. This aggregated view prevents scope creep and maintains focus on the connections between metrics.
>
>
> **W3 & Q2: Codebase behavior and CNN-to-Linear stitching.**
>
> The script returns zero for CNN-to-Linear stitching to handle structural incompatibilities gracefully and prevent execution errors during automated runs.
>
> This does not affect the aggregated similarity values reported in the paper, as we intentionally excluded cross-architecture stitching (e.g., CNN-to-Linear) from our analysis. All pairwise comparisons in our results are strictly intra-architecture (CNN-to-CNN and MLP-to-MLP) to isolate representational differences from raw structural incompatibilities. We will clarify this design choice in line 298 of the revised manuscript.
>
>
> **W4: Usable information as a "rephrasing" of linear correlation.**
>
> We disagree that our measure reduces to linear correlation, as that assumes "coordinate-wise correspondence." Because independent neural networks have distributed representations and misaligned bases, dimension-by-dimension correlation fails. Instead, our framework uses usable information to optimize over a transformation family V for global alignment, a process mathematically richer than computing static coordinate-wise covariance.
>
> Most importantly, our main contribution is not a new metric, but using usable information as a tool to unify historically disconnected similarity paradigms (functional vs. representational) under a single framework.
>
>
> **Q1: Sign of the accuracy-ratio difference in Figure 2a.**
>
> Correct. The sign merely reflects comparison direction; absolute difference is more informative. We will update Figure 2a to show absolute values in the final version.
>
>
> **Q3: Convolutional representations and similarity metrics.**
>
> We thank the reviewer for this point; it is an important distinction we overlooked, and we will add a comment on it in the final version.
>
> The comparisons in Figure 3 are restricted to MLP architectures. From a geometric perspective, metrics like CKA and RSA are spatially agnostic, whereas a convolutional stitcher $\psi$ explicitly preserves local 2D structures. Comparing these would be inconsistent, as CKA/RSA would collapse the very spatial geometry that $\psi$ is designed to leverage. To maintain a mathematically consistent geometric space for evaluating the connections between metrics, we focused this analysis on MLPs.
>
>
> **Q4: Non-deterministic tasks.**
>
> We assume $Y = f(X)$ because this directly reflects standard machine learning datasets. In practice, datasets almost universally consist of fixed $\{x, y\}$ pairs, where a specific input $x$ is consistently mapped to a single, deterministic label $y$. Our theoretical statements are explicitly designed to model this standard empirical setting.

---

> > ### Author Rebuttal · Reviewer_pfar · 2026-04-03
> >
> > Unresolved:
> > - **W1**: The proposed rephrasings/reformulations do not meaningfully close the gap between assumptions of perfect equivalence (misleadingly referred to as functional/representational *similarity*) and the imperfect equivalence relevant to practice.
> > - "The inherent directional asymmetry of model stitching has not been previously identified or formally discussed."  First, calling the directional asymmetry *inherent* inflates the claim past the empirical observation that is presented in this work.  Second, Bansal 2021 pointed out asymmetry in stitching (see the subsection labeled *Asymmetry*), noting the benefit over symmetric similarity measures like CKA.  While there is value in the empirical findings of the current work, its impact should not be overstated.
> > - Linear-CNN comparisons: As written, the paper gives the impression of exhaustive pairwise comparison across models and layers. (L277: *"we train several models across multiple datasets and compare all intermediate layers of every model pair with each other."*) If unsupported cross-architecture comparisons were in fact excluded, the current description is misleading. If they were not excluded cleanly in the analysis pipeline, then some reported trends may partly reflect implementation artifacts rather than representational phenomena.
> > - Rebuttal states CNNs were excluded from RSA/CKA/SVCCA results (Fig 3): As written, the paper fails to specify key aspects of the empirical analysis, and gives the impression that comparisons were exhaustive.
> >
> > At large, my concerns about significance remain.  New details about the analysis posted in the rebuttal point to underspecification of key information.

---

> > > ### Author Response · Authors · 2026-04-04
> > >
> > > We sincerely thank the reviewer for the continued engagement. You raise highly valid points regarding the framing of our claims and the precision of our methodological descriptions. We appreciate you catching these oversights, and we address your remaining concerns below:
> > >
> > > **Perfect vs. Imperfect Equivalence**
> > >
> > > We agree with the reviewer's distinction. In our theoretical proofs, we will replace the term "similarity" with "equivalence" (e.g., Functional and Representational Equivalence), as this term mathematically reflects the idealized regime much more adequately. We will introduce the term "similarity" only in the final metrics section to explicitly connect our formal bounds to the imperfect, continuous regime evaluated in our experiments.
> > >
> > > ---
> > > **Stitching Asymmetry & Bansal 2021**
> > >
> > > We thank the reviewer for highlighting Bansal 2021. We will update the text to explicitly credit them for previously pointing out this phenomenon. However, to the best of our knowledge, this asymmetry had not been previously formalized nor deeply studied.
> > >
> > > Consequently, we respectfully push back on the assertion that our claim of *inherent* asymmetry inflates past our empirical observations. In Proposition 3.5, we formally prove that $Z_1$ is perfectly stitchable into $Z_2$ if and only if $Z_1$ is a Markov blanket for $Y$ relative to $Z_2$. Because this Markov blanket condition is strictly directional when $Z_1 \neq Z_2$, the asymmetry of the stitching transformation is a mathematical inevitability in general cases, not just an empirical artifact. We will revise the manuscript to clearly delineate prior empirical observations from our new theoretical formalization and comprehensive empirical analysis.
> > >
> > > ---
> > > **Underspecification of Empirical Analysis (Linear-CNN & Fig 3)**
> > >
> > > We completely agree that our phrasing at L277 ("compare all intermediate layers of every model pair") is overly broad and gives a misleading impression of exhaustive cross-architecture comparisons. We want to categorically reassure the reviewer that our data pipeline strictly and cleanly filtered out cross-architecture comparisons; there are no implementation artifacts or execution errors polluting the reported trends.
> > >
> > > We acknowledge that failing to specify this in the text, as well as omitting the explicit restriction of Figure 3 to MLP architectures, was an oversight that undersells the rigor of our experimental design. Although we cannot modify the PDF during this specific phase of the rebuttal process, we guarantee that the final version will be extensively updated to explicitly detail these exclusions and correct the methodological underspecification.

---

### Decision · Program_Chairs · 2026-04-30

**Decision:**

Accept (regular)

**Comment:**

## Summary of paper

This paper proposes a unified framework connecting functional similarity (whether two representations support the same downstream tasks) and representational similarity (whether they encode the same information about the input) through usable information theory. CKA and RSA are reinterpreted as estimators of usable information under specific transformation constraints. Experiments compare pairwise intermediate layers across MLPs, CNNs, and modern architectures on five small-image datasets.

## Reviews and discussions

The reviews lean positive: JoBF (5), dfiG (4), bjPf (4), and pfar (3).

Pro arguments (among others):
- unification of functional and representational similarity through usable information is novel and valuable (all)
- the stitching asymmetry result is practically relevant (all)
- CKA and RSA reinterpreted as estimators of usable information (bjPf, dfiG)
- good presentation (all)

Con arguments (among others):
- gap between exact-equivalence theory (conditional MI = 0) and the approximate similarity regime relevant in practice (pfar, dfiG)
- overclaiming relative to what is proved vs empirically observed (pfar, dfiG) -- authors agreed to tone down
- soundness of proofs: Prop 3.5 and 4.2 and Sec 4 derivations (dfiG) -- authors acknowledged

## AC's assessment

The AC has read the paper and all reviews in detail.

The concerns about the gap between exact-equivalence theory and practical approximate similarity is valid. The theoretical results hold in an idealised regime, and the bridge to continuous metrics is empirical. The original submission did overclaim on certain arguments. In response, the authors have agreed to adjust the scope and tone. These are reasonable fixes for the camera-ready.

Despite these concerns, the AC believes the paper should be accepted. The core conceptual contribution of unifying functional and representational similarity is valid and is appreciated by the community. At the very least, the paper provides a foundation that future work can refine.

The AC expects the camera-ready to deliver on the promised revisions.

## Recommendation: accept